evolution, theoretical biology, behaviour

extra-pair paternity, mate guarding, alternative reproductive tactics, sexual conflict, tragedy of the commons, evolutionary game theory

**Author for correspondence:**
Xiang-Yi Li
e-mail: li@evolbio.mpg.de

# Coevolution of female fidelity and male help in populations with alternative reproductive tactics

Xiang-Yi Li[1], Andrew Morozov[2,3] and Wolfgang Goymann[4]

[1]Institute of Biology, University of Neuchâtel, Rue Emile-Argand 11, 2000 Neuchâtel, Switzerland
[2]Department of Mathematics, University of Leicester, University Road, Leicester LE1 7RH, UK
[3]Institute of Ecology and Evolution, Russian Academy of Sciences, Moscow, Russia
[4]Department of Behavioural Neurobiology, Max Planck Institute for Ornithology, Eberhard-Gwinner-Straße, 82319, Germany

 X-YL, 0000-0001-8662-0865; AM, 0000-0002-6935-3563; WG, 0000-0002-7553-5910

In socially monogamous species, pair-bonded males often continue to provide care to all offspring in their nests despite some degree of paternity loss due to female extra-pair copulation. Previous theoretical models suggested that females can use their within-pair offspring as 'hostages' to blackmail their social mates, so that they continue to provide care to the brood at low levels of cuckoldry. These models, however, rely on the assumption of sufficiently accurate male detection of cuckoldry and the reduction of parental effort in case of suspicion. Therefore, they cannot explain the abundant cases where cuckolded males continue to provide extensive care to the brood. Here we use an analytical population genetics model and an individual-based simulation model to explore the coevolution of female fidelity and male help in populations with two genetically determined alternative reproductive tactics (ARTs): sneakers that achieve paternity solely via extra-pair copulations and bourgeois that form a mating pair and spend some efforts in brood care. We show that when the efficiency of mate guarding is intermediate, the bourgeois males can evolve to 'specialize' in providing care by spending more than 90% of time in helping their females while guarding them as much as possible, despite frequent cuckoldry by the sneakers. We also show that when sneakers have tactic-specific adaptations and thus are more competitive than the bourgeois in gaining extra-pair fertilizations, the frequency of sneakers and the degrees of female fidelity and male help can fluctuate in evolutionary cycles. Our theoretical predictions highlight the need for further empirical tests in species with ARTs.

## 1. Introduction

Apart from a few exceptions [1–4], socially monogamous females and males often mate multiply and produce extra-pair offspring (EPO) [5,6]. Female promiscuity often leads to (sometimes substantial) paternity loss to the care-providing social male. For example, in the cichlid fish *Variabilichromis moori*, paired brood-tending males lose on average 37% of paternity in their broods, while siring very few EPO themselves [7]. Extra-pair paternity was also found in 70% of the nests of the socially monogamous beetle *Odontotaenius disjunctus*, where 54.8% of the offspring were extra-pair [8]. Many socially monogamous birds with biparental care also have surprisingly high proportions of EPO in their broods, such as black redstarts (*Phoenicurus ochruros*, 30.2% of all broods and 28.8% offspring) [9], Magellanic penguins (*Spheniscus magellanicus*, 48% of all broods and 31% of offspring) [10] and tree swallows (*Tachycineta bicolor*, 75% of all broods and 51% of offspring) [11].

Why do the cuckolded males continue to provide paternal care? A previous model [12] showed that males could be 'blackmailed' by their paired mates to continue providing care, because otherwise their own genetic offspring will also suffer. Everything else being equal, a male tolerating some degree of cuckoldry may have higher fitness than a male who stops caring altogether. This model requires the simultaneous presence of several conditions, including reasonably accurate detection of cuckoldry by males and the reduction of parental effort in case of suspicion. Empirical work in birds (e.g. western bluebirds *Sialia Mexicana* [13], black redstarts [9], scarlet rosefinches *Carpodacus erythrinus* [14] and azure-winged magpies *Cyanopica cyanus* [15]), fishes (e.g. the plainfin midshipman *Porichthys notatus* [16], and the cichlid *Variabilichromis moori* [7]) and the burying beetle (*Nicrophorus vespilloides*) [17], however, found that males do not seem to detect cuckoldry or react to it [for instance, in western bluebirds [13], males did not reduce paternal effort even when they observed their mate engaging in extra-pair copulations (EPCs)]. We aim to find out how such apparently maladaptive male investment can evolve.

Pair-bonded males, however, are not always 'reticent victims'. They may guard their mates to actively prevent paternity loss. Models [18,19] have shown that male mate-guarding can be effective in assuring paternity. For example, in the model of Fishman *et al.* [18], females are predicted to always seek EPCs, while a male either spends all his time attempting EPCs, or guards his female during her fertile period. The model of Kokko & Morrell [19], on the other hand, assumed females to be fertile synchronously or that males cannot detect whether a female is currently fertile or not. Since males face the trade-off between achieving paternity 'at home' and elsewhere, the intensity of mate-guarding has a nonlinear relationship with female fidelity—with males guarding most intensely when female fidelity is intermediate [19].

The ESS approach used in these previous models, however, only predicts the overall levels of male help or guarding in the population, but remains silent about how these are realized through individual behavior. Those models, therefore, implicitly assumed a monomorphic population of males where everyone uses the same tactic, despite the fact that male alternative reproductive tactics (ARTs) are widespread in nature [20]. On the other hand, there is also a large body of modelling work on the competition and maintenance of male ARTs, including Parker's seminal 'sneaks and guarders' model [21,22], focusing on sperm competition and its evolutionary consequences. Most of the ARTs models were constructed based on evolutionary games such as modified producer–scrounger or hawk–dove games and also relied on the ESS approach [23–29]. These models typically focus on the competition between different male tactics while ignoring the interactions with females. Even in those that explicitly considered the female part of the population [26,27], the fecundity of females was unaffected by the phenotypes of males that they interacted with.

We aim to bridge the knowledge gap by studying the coevolution of male help and female fidelity at the presence of ARTs, without assuming that males can detect cuckoldry. In our models, we consider two male ARTs, namely, 'sneakers' that attempt to achieve paternity solely through EPCs, and 'bourgeois' that attempt to form social pairs with one female and spend some time caring for the brood (following the nomenclature in [30]). We assume that males cannot

detect whether a female is in her fertile period or not [19]. The two ARTs are assumed to be genetically determined. We are aware that this simplification ignores the fact that in nature ARTs may depend on condition or on complex gene-by-environment interactions [31–33]. The main reason for adopting this simplified assumption was to provide a null expectation of the evolutionary dynamics, so that our model can serve as a basis for future models that include further complexities.

We first use an analytical population genetics model to map the population fitness landscape under different combinations of female fidelity and male help values. We then use individual-based simulations to study the scenarios where the degree of either female fidelity or male help is allowed to evolve while the other is fixed, and the coevolution of both. We show that although cooperation between males and females (i.e. high values of female fidelity and male help) should lead to the highest population growth rate, sexual conflict and the male–male competition can trap both sexes into a 'tragedy of the commons', allowing sneakers to invade, and reducing population growth rate. We also show that when the efficiency of mate guarding is intermediate, bourgeois males may evolve to investing almost all their time in paternal care, despite severe cuckoldry by the sneakers. Furthermore, if the sneakers are more competitive than the bourgeois males in extra-pair fertilizations, the degrees of female fidelity, male help and the frequency of sneakers can fluctuate in evolutionary cycles.

## 2. Models

### (a) The analytical population genetics model

We assume that the sneaker and bourgeois male ARTs are genetically determined by an autosomal locus with two alleles $A$ and $a$. Males with the $AA$ and $Aa$ genotypes are bourgeois, and those with the $aa$ genotype are sneakers. The numbers of females and males with different genotypes are represented by $F_i$ and $M_i$ ($i = AA, Aa, aa$). Social pairs are formed by females and bourgeois males. The number of social pairs ($P$) is therefore

$$P = \text{minimum } (F_{AA} + F_{Aa} + F_{aa}, M_{AA} + M_{Aa}). \quad (2.1)$$

We introduce the following parameters of life-history traits. The fidelity of females, $u \in [0, 1]$, represents the proportion of time a female spends without the intention for EPC (assuming that the EPC-seeking males cannot force her to mate). The degree of male help, $h \in [0, 1]$, represents the proportion of time spent by the social male on brood care. The survival rate of within-pair offspring (WPO) is an increasing function of male care, $S(h) = \sqrt{h}$. The EPO may have a survival advantage or disadvantage relative to WPO. In the absence of individual quality variation, the different survival rates between WPO and EPO may be caused by maternal effects [34–37] or genetic compatibility. For example, in blue tits (*Cyanistes caeruleus*) [38,39], dark-eyed juncos (*Junco hyemalis*) [40] and reed buntings (*Emberiza schoeniclus*) [41], EPO survive better due to increased heterozygosity, while in song sparrows (*Melospiza melodia*) [42,43] and house sparrows (*Passer domesticus*) [44], EPO were less likely to survive, probably due to inbreeding depression [45]. We denote the relative survival rate of EPO as $r$ ($r > 0$), so that their survival rate is $S(h)r$. Because males that provide care almost always guard their mates to some extent, we model mate-guarding as a

by-product of brood care with an efficiency $\delta \in [0, 1]$. With probability $\delta$, the social male can successfully prevent his female from EPCs given that he is currently helping the female at home. Using the above notions and assuming that within- and extra-pair matings are equally likely to fertilize an egg, the expected proportion of EPO females produce is $E = (1 - u)(1 - \delta h)$. The number of offspring each paired female can produce is $R$, and the sex ratio at birth is $\lambda$ (proportion of males). In the current model, we assume that $u$, $h$, $r$, $\delta$, $R$ and $\lambda$ are constants.

Under random pair-formation, the proportion of the social pairs of type $ij$ (where $i = AA$, $Aa$, $aa$ is the type of female and $j = AA$, $Aa$ is the type of male) is given by $F_i M_j / (F_{AA} + F_{Aa} + F_{aa})(M_{AA} + M_{Aa})$. Each type of social pair $ij$ produces offspring with genotypes following the Mendelian law of inheritance. Reproduction happens discretely at the end of each breeding season. Denote $\theta = P(1 - \lambda)(1 - E) RS(h)/(F_{AA} + F_{Aa} + F_{aa})(M_{AA} + M_{Aa})$, the

$$
\begin{aligned}
\widetilde{M_{AA}} &= P\frac{M_{AA}}{M_{AA} + M_{Aa}}(1 - h) + \left[M_{AA} - P\frac{M_{AA}}{M_{AA} + M_{Aa}}\right] = -P\frac{M_{AA}}{M_{AA} + M_{Aa}}h + M_{AA}; \\
\widetilde{M_{Aa}} &= P\frac{M_{Aa}}{M_{AA} + M_{Aa}}(1 - h) + \left[M_{Aa} - P\frac{M_{Aa}}{M_{AA} + M_{Aa}}\right] = -P\frac{M_{Aa}}{M_{AA} + M_{Aa}}h + M_{Aa}
\end{aligned}
\right\}
\tag{2.3}
$$
$$\text{and} \quad \widetilde{M_{aa}} = M_{aa}.$$

In (2.3), the terms in the square brackets describe the numbers of bourgeois males that were unable to form a social pair (e.g. due to a male-biased sex ratio). In this case, they behave like sneakers and spend all their time seeking EPCs. Assuming that EPCs occur randomly between males and females that are currently seeking opportunities, the proportions of each genotype combination are

$$
\frac{\widetilde{M_i} F_j}{(F_{AA} + F_{Aa} + F_{aa})(M_{AA} + M_{Aa} + M_{aa} - h P)}, \quad i = AA, Aa,
$$
$$j = AA, Aa, aa;$$

and

$$
\frac{M_{aa} F_j}{(F_{AA} + F_{Aa} + F_{aa})(M_{AA} + M_{Aa} + M_{aa} - h P)}, \quad j = AA, Aa, aa.
$$

Denoting $\psi = P(1 - \lambda)E RS(h)r/(F_{AA} + F_{Aa} + F_{aa})(M_{AA} + M_{Aa} + M_{aa} - h P)$ and applying the Mendelian law of inheritance, the number of survived of female EPO of each genotype produced over a generation are

$$
\begin{aligned}
\Delta_{ep}F_{AA} &= \psi\left(F_{AA}\widetilde{M_{AA}} + \frac{1}{2}F_{AA}\widetilde{M_{Aa}} + \frac{1}{4}F_{Aa}\widetilde{M_{Aa}} + \frac{1}{2}F_{Aa}\widetilde{M_{AA}}\right); \\
\Delta_{ep}F_{Aa} &= \psi\left(\frac{1}{2}F_{AA}\widetilde{M_{Aa}} + \frac{1}{2}F_{Aa}\widetilde{M_{Aa}} + \frac{1}{2}F_{Aa}\widetilde{M_{AA}} \right. \\
&\quad \left. + F_{aa}\widetilde{M_{AA}} + \frac{1}{2}F_{aa}\widetilde{M_{Aa}} + F_{AA}M_{aa} + \frac{1}{2}F_{Aa}M_{aa}\right) \\
\text{and} \quad \Delta_{ep}F_{aa} &= \psi\left(\frac{1}{4}F_{Aa}\widetilde{M_{Aa}} + \frac{1}{2}F_{aa}\widetilde{M_{Aa}} + \frac{1}{2}F_{Aa}M_{aa} + F_{aa}M_{aa}\right).
\end{aligned}
\right\}
\tag{2.4}
$$

As previously, the production of male EPO of the corresponding genotype is described by replacing the term $(1 - \lambda)$ in $\psi$ with $\lambda$.

numbers of survived female WPO of each genotype are

$$
\begin{aligned}
\Delta_{wp}F_{AA} &= \theta\left(F_{AA}M_{AA} + \frac{1}{2}F_{AA}M_{Aa} + \frac{1}{4}F_{Aa}M_{Aa} + \frac{1}{2}F_{Aa}M_{AA}\right) \\
\Delta_{wp}F_{Aa} &= \theta\left(\frac{1}{2}F_{AA}M_{Aa} + \frac{1}{2}F_{Aa}M_{Aa} + \frac{1}{2}F_{Aa}M_{AA} + F_{aa}M_{AA} \right. \\
&\quad \left. + \frac{1}{2}F_{aa}M_{Aa}\right) \\
\text{and} \quad \Delta_{wp}F_{aa} &= \theta\left(\frac{1}{4}F_{Aa}M_{Aa} + \frac{1}{2}F_{aa}M_{Aa}\right),
\end{aligned}
\right\}
\tag{2.2}
$$

The number of survived male WPO of each genotype produced within a generation is described by similar expressions where the term $1 - \lambda$ in $\theta$ is replaced with $\lambda$.

Besides the WPO, females also produce offspring via EPCs. The number of males of each genotype that are currently seeking EPCs are

Now we describe the population dynamics. We consider that the population contains overlapping generations and has a fixed carrying capacity $N_0$. If at the end of a reproductive season the population size exceeds $N_0$, individuals of each genotype are culled proportional to their abundance.

The total number of individuals $N(t)$ at the end of generation $t$ is

$$
\begin{aligned}
N(t) &= \sum (F_i(t) + M_i(t) + \Delta_{wp}F_i(t) + \Delta_{wp}M_i(t) + \Delta_{ep}F_i(t) \\
&\quad + \Delta_{ep}M_i(t)), \\
i &= AA, Aa, aa.
\end{aligned}
\tag{2.5}
$$

The number of females at the start of generation $t + 1$ is given by

$$
\left.
\begin{aligned}
F_i(t + 1) &= F_i(t) + \Delta_{wp}F_i(t) + \Delta_{ep}F_i(t), \text{ for } N(t) < N_0 \\
\text{and} \quad F_i(t + 1) &= \frac{(F_i(t) + \Delta_{wp}F_i(t) + \Delta_{ep}F_i(t))N_0}{N(t)}, \text{ for } N(t) > N_0.
\end{aligned}
\right\}
\tag{2.6}
$$

The dynamics for males can be obtained in similar ways.

When female fecundity $R$ is sufficiently high, the population size can stay at $N_0$ through generations. We are interested in finding the equilibrium proportion of sneakers in the population, and the *per capita* population growth rate within a generation

$$
\begin{aligned}
W_t &= \frac{\sum (\Delta_{wp}F_i(t) + \Delta_{wp}M_i(t) + \Delta_{ep}F_i(t) + \Delta_{ep}M_i(t))}{\sum (F_i(t) + M_i(t))}, \\
i &= AA, Aa, aa, \text{ for } t \to \infty.
\end{aligned}
\tag{2.7}
$$

We use $W_t$ to describe the fitness landscape of the population at different combinations of female fidelity and male help values.

## (b) Individual-based simulations

It is possible to extend the above analytical framework to allow male help ($h$) and female fidelity ($u$) to evolve and to include additional biological factors and processes, but the expressions quickly become overly cumbersome and difficult to analyse. To circumvent these difficulties, we built an individual-based simulation model with explicit 'genetic architectures' associated with different traits. The simulation model shares most of the assumptions of the analytical model, and therefore we only describe the differences below.

The first important difference is that we allow the degrees of female fidelity ($u$) and male help ($h$) to evolve. Each individual now has three evolving diploid autosomal loci, including (1) an ART locus that is only expressed in males and determines whether the male is a sneaker or a bourgeois, (2) a fidelity locus that is only expressed in females, (3) a male help locus that is only expressed in males. The ART locus has two discrete alleles $A$ and $a$, same as in the analytical model. It evolves by changing the relative frequencies of the two alleles. The other two loci have alleles that take continuous values between 0 and 1, and evolve as quantitative traits. The phenotype of an individual determined by the loci with continuous values is the mean of the two allelic values. All loci are subject to Mendelian inheritance without linkage. Unless otherwise stated, the mutation rate at each locus is set to 0.01. For the loci with continuous allelic values, the magnitude of the mutations follows a normal distribution with zero mean and a standard deviation of 0.01. The sex of each offspring is determined randomly with $1:1$ sex ratio.

The second crucial difference is that we now allow the possibility for sneakers to have some additional advantage in extrapair fertilization (i.e. due to tactic-specific adaptations, such as allocating more resources into sperm production, as predicted by theory [21] and shown by empirical work [46]). In the above analytical model, the difference in extra-pair fertilization success between sneakers and paired bourgeois was simply proportional to the time they spent on seeking EPCs (i.e. paired bourgeois : sneaker $= (1 - h) : 1$. Now we allow the advantage of sneakers to exaggerate, so that the success ratio between paired bourgeois to sneakers becomes $(1 - h)^{\beta} : 1$, where $\beta > 1$.

The last important difference is that we assumed overlapping generations in the population genetics model but nonoverlapping generations in the individual-based simulations. We show in the electronic supplementary material, section B that modifying the population genetics model to non-overlapping generations produces qualitatively the same results.

Here we only present the results under the simplest scenarios where individual variation in quality is absent, sex ratio is fixed at 0.5 throughout the life cycle, random pair formation, and random encounters between EPC-seeking males and females. The annotated simulation codes provided on the GitHub repository include additional options to explore the effects of more complex genetic structures that introduce variations in individual quality, biased sex ratios and varying intensities of intrasexual competition.

## 3. Results

## (a) The population fitness landscapes

We first present the population fitness landscape generated by the analytical model [represented by the *per capita* growth rate, equation (2.7)] under different combinations of female fidelity and male help values (figure 1). It is intuitive to understand that when EPO have lower survival rates (figure 1, $r = 0.9$), both sexes are more interested in producing WPO, and the growth rate of the population is the highest when females are fully loyal and when males spend all their time providing care. But what happens when EPO have a survival advantage ($r > 1$)? Does the population grow faster at smaller fidelity($u$) values then? We show that when $r$ is not too large (figure 1, $r = 1.1$), the population fitness landscape still peaks at $u = h = 1$. This is because as female fidelity decreases, the frequency of sneakers increases (see electronic supplementary material, figure S1), causing the frequency of care-giving bourgeois to decrease. Because the sex ratio is balanced and a female can reproduce only if paired with a bourgeois male, the number of reproducing females decreases. Therefore, when $r$ is not too large, the survival benefit of EPO does not compensate for the reduced number of reproducing females. Only when $r$ is very large (around 3, probably beyond a biologically realistic range), the peak of the population fitness landscape was shifted to lower $u$ and $h$ values.

The topography of the population fitness landscape, however, does not always predict the trajectory of evolution. Sexual conflict and intrasexual competition can often cause the breakdown of cooperation between individuals, leading to a 'tragedy of the commons'. We will show in the following with individual-based simulations that this can happen when EPO have a survival advantage ($r > 1$).

## (b) When only male help or female fidelity can evolve

From here onwards, we present results generated from the simulation model. Before studying the complete coevolutionary dynamics, we first examine the simpler cases where either the degree of male help or female fidelity is fixed, and only the other can evolve. These correspond to the biological scenarios where the standing genetic variation at one of the loci is much more abundant than at the other locus. We show in figure 2 the equilibrium population states when EPO have a survival advantage relative to WPO ($r = 1.1$) so that females have an incentive to cheat. The corresponding case where the sneakers are prevented from evolving (by setting the initial frequency of $a$ allele to 0 and turning off mutations at this locus) is provided in electronic supplementary material, figure S12. When the degree of male help is fixed (figure 2a), the frequency of EPO is high except when mate guarding is efficient (high $\delta$) and bourgeois males spend high proportions of time providing care while guarding their females (high $h$). Without either of the two conditions, females cannot be prevented from frequent cheating. When mate guarding is inefficient (small $\delta$) but the fixed degrees of male help is high (high $h$), paired bourgeois males provide high levels of care to the brood but suffer tremendously from cuckoldry. In this case, the sneakers can freeride the brood care provided by the bourgeois males and reach high frequency. When the degree of male help is fixed and EPO have a survival advantage, females always evolve to relatively low fidelity.

When female fidelity is fixed (figure 2b), as long as they are relatively loyal (large $u$), the help provided by their social mates can pay off, and the degree of male help evolve to high values especially when mate guarding is efficient (high $\delta$). But when female fidelity is fixed to low values, they cannot be prevented

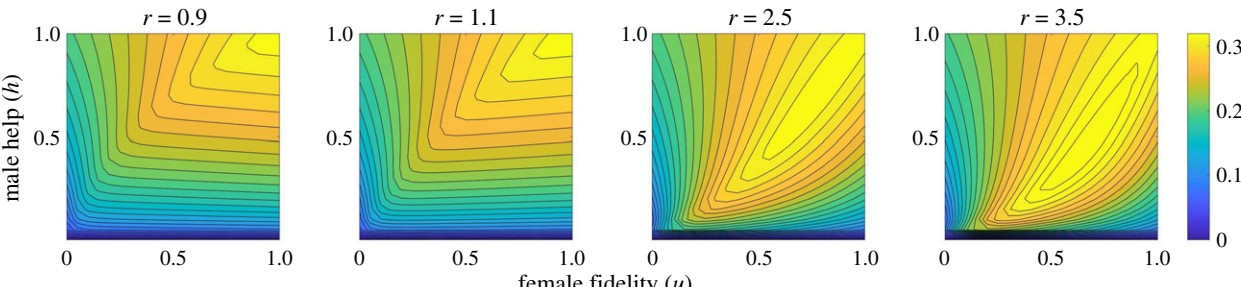

**Figure 1.** The population fitness landscape, represented by the *per capita* growth rate of the population at the evolutionary equilibrium, under different *r* and different levels of *u* and *h* combinations. The efficiency of mate guarding $\delta = 0.5$. The corresponding results with qualitatively similar pattern for $\delta = 0.1$ and $\delta = 0.9$ are provided in electronic supplementary material, figures S2 and S4. The corresponding qualitatively similar results produced from the modified non-overlapping generation model are provided in the electronic supplementary material, section B. (Online version in colour.)

from cheating if mate guarding is inefficient, and consequently, the equilibrium frequency of EPO is extremely high in this parameter region. Because caring for a brood filled with EPO does not pay off for the bourgeois, male help evolves to low values. The frequency of sneakers is, however, not particularly high in this region (low *u* and low *δ*), because like the sneakers, the paired bourgeois males also spend almost all their time attempting EPC, and thus the sneakers do not have much competitive advantage. Interestingly, when females are of low fidelity but the efficiency of mate guarding is intermediate, the male population evolves to 'specialize' in different tactics with the bourgeois investing around 90% of their time caring for the brood despite high levels of cuckoldry (approx. 40% of the offspring are extra-pair), with a large proportion of sneakers in the population. When mate guarding is very efficient (in the extreme case, $\delta = 1$), females have no chance for EPC and therefore male help evolves to close to 1, sneakers are excluded, and there are almost no EPO in the population.

### (c) When female fidelity and male help coevolve

After studying the evolutionary dynamics of either male help or female fidelity while the other is fixed, we allowed the two traits to coevolve and observed the population state at equilibrium. We compared the results under different survival rates of EPO (*r* ranges from 0.8 to 1.2) with that in a population without sneakers (by setting the initial frequency of the *a* allele to 0 and turning off mutations at this locus). As shown in figure 3, the results were similar except in the region where the efficiency of mate-guarding is intermediate (*δ* ranges between 0.4 and 0.9) and the EPO have a survival advantage (*r* > 1). In this region, the bourgeois male evolved to providing extensive care while trying to prevent cuckoldry as much as possible. The end result is fewer EPO in the population, higher levels of male care and overall survival of offspring, despite some unpreventable cuckoldry by the sneakers.

No matter whether sneakers were present or not, the population evolved to a 'tragedy of the commons' when EPO have a survival advantage (*r* > 1 so females have an incentive to cheat) and mate guarding is very inefficient (*δ* < 0.4, where the paired bourgeois can hardly prevent their females from cheating). In this region, females evolve to be highly disloyal, the males do not help with brood care, and most offspring produced are EPO. In the simulations, male help did not reach zero only because of mutation–selection balance, and we had to assign females very high baseline fecundity so that the population is sustainable (see electronic supplementary material,

figure S16 and S17 for sample evolutionary trajectories). Under natural conditions, such populations are likely to go extinct. Only when WPO survive better than EPO (*r* < 1), the population evolves towards the peak of the population fitness landscape where the bourgeois are highly helpful, females highly loyal, sneakers can hardly invade the population, and almost all offspring produced are WPO. When *r* = 1, females do not evolve to increase or decrease their fidelity because WPO and EPO are equally valuable to them. The frequency of EPO decreases and the degree of male help increases as the efficiency of mate guarding increases, and the parameter region where males evolve to very high levels of help (*h* > 0.9) is larger when sneakers are present. Interestingly, when mate guarding is highly efficient (*δ* is close or equal to 1), female fidelity does not evolve to increase or decrease but fluctuates around 0.5 under random genetic drift, no matter whether EPO survive better or worse than WPO. This is because when the bourgeois can efficiently prevent their females from cheating, the (genetically determined) intrinsic fidelity of the females does not matter anymore.

### (d) When sneakers have an advantage in extra-pair fertilization

So far, the advantage of sneakers relative to the bourgeois in extra-pair fertilization has been proportional to the time they spend on seeking opportunities (i.e. sneakers spend full time while the bourgeois only a proportion of $1 - h$). In this sense, the sneakers were effectively the same as bourgeois with the allelic value at the *h* locus fixed to zero. The coevolutionary dynamics always evolved towards an equilibrium and then fluctuated around it. But what happens if the sneakers are better at gaining extra-pair fertilization opportunities due to strategy-specific adaptations? As shown in figure 4, by scaling the time spent on attempting EPC between a bourgeois and a sneaker with a factor $\beta > 1$ so that their relative success of EPC fertilization becomes $(1 - h)^{\beta} : 1$, the population state can evolve in cycles. Larger values of $\beta$ leads to more frequent cycles (electronic supplementary material, figure S14). More sample trajectories are provided in electronic supplementary material, figures S13 and S14.

The cycles can emerge even when WPO and EPO have the same survival rates and when mate-guarding is totally inefficient (*r* = 1 and *δ* = 0 in figure 4). Within a cycle, when both male help and female fidelity are very high, a female mutant with lower fidelity has a fitness advantage by mating more

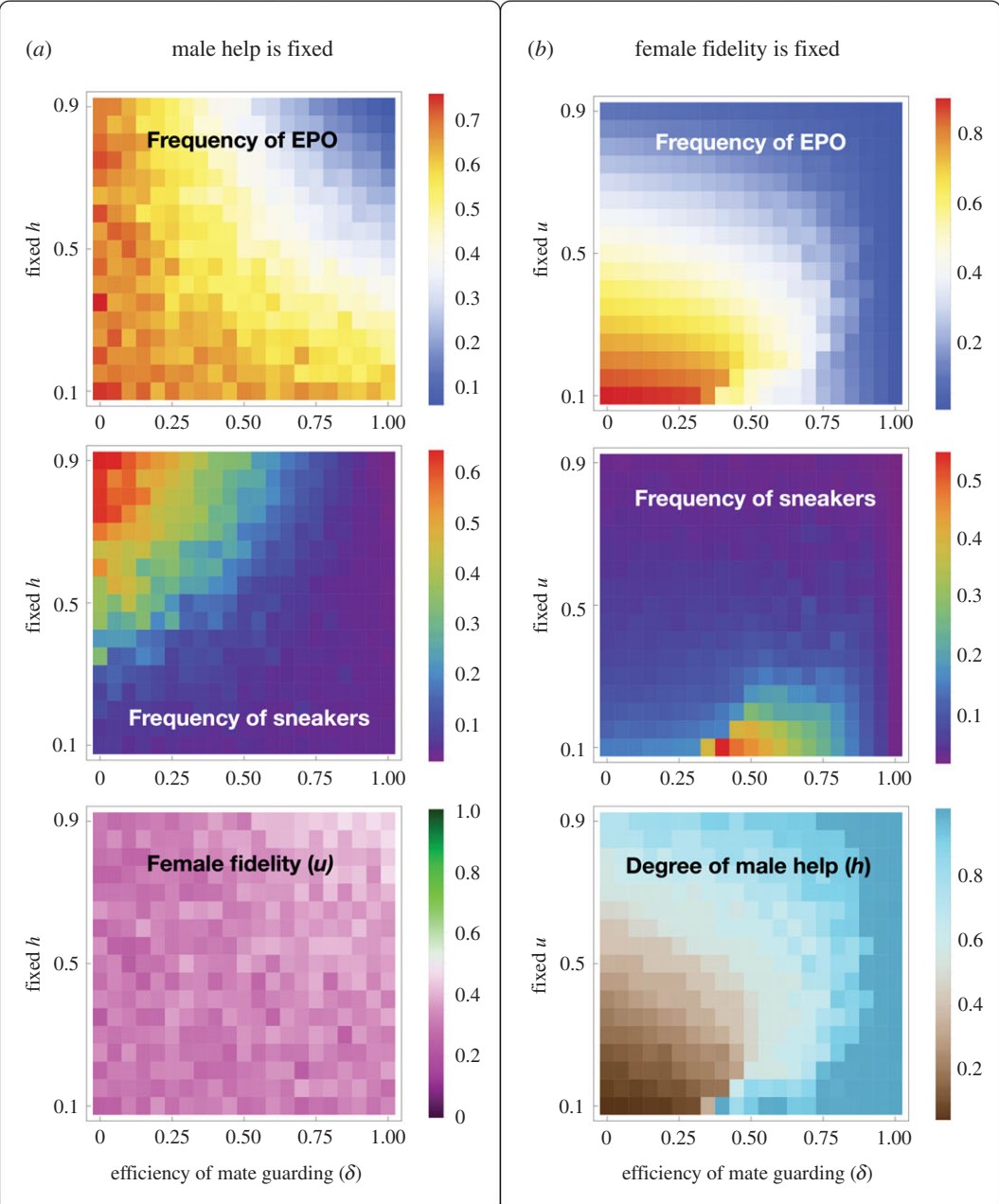

**Figure 2.** Heatmaps of the population states at the evolutionary equilibrium when either male help is fixed while the degree of female fidelity can evolve (left column) or female fidelity is fixed while the degree of male help can evolve (right column). The data at the same pixel position across the heatmaps in (*a*) are generated from the same simulation run. The same is for the heatmaps in (*b*). Each simulation was run for 5000 and 15 000 generations when *h* is fixed and when *u* is fixed, respectively, because the former simulations reach equilibrium faster. The value at each pixel is the mean of the last 500 generations. In each simulation, $r = 1.1$, population size was 5000, the initial frequency of the *a* allele was 0.05. The initial *h* was 0.5 in the left column, and the initial *u* was 0.5 in the right column. (Online version in colour.)

often with sneakers and thereby producing more sneaker sons that free-ride the paternal care efforts of the bourgeois males, and producing daughters that also have low fidelity and mate often with the sneakers. The positive feedback resembles the Fisherian 'sexy sons' mechanism in the sense that low fidelity females mate more often with sneakers (as if it was their preference) and produce 'sexy (sneaker) sons' and daughters also of low fidelity (sharing the same preference for sneakers). This leads to a rapid increase of sneaker frequency and a sharp drop of female fidelity in the population. Because male help by the bourgeois does not pay off any more, it also drops quickly. The short-term success of sneakers is however self-defeating because of negative frequency-dependent selection. When there are too many sneakers in the population,

females of higher fidelity have an advantage because they produce relatively more bourgeois sons and fewer sneakers (who now suffer from over competition and low fitness), and their daughters also mate more often with the bourgeois males. Again, the process resembles the Fisherian 'sexy sons' mechanism, with the preferred male type switched to the bourgeois. The increased fidelity of females further leads the bourgeois males to provide more paternal care. This process drives the frequency of sneakers to further decrease, and the fidelity of females and the help of bourgeois males to further increase, until the next cycle starts. Note that the cycles disappear when sneakers are absent (see electronic supplementary materials, section E for sample trajectories and explanations).

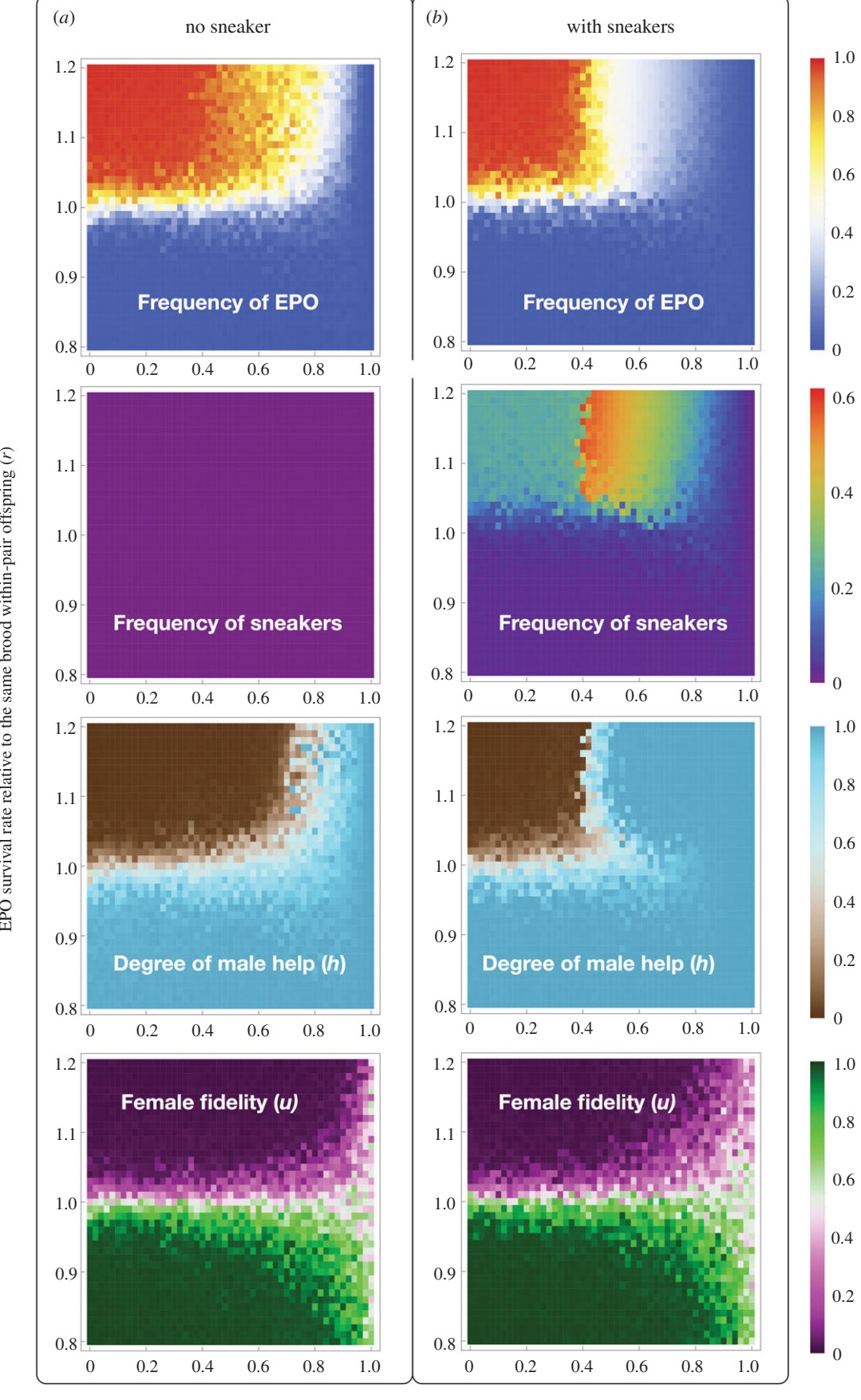

**Figure 3.** Heatmaps of the equilibrium population state under different $r$ and $\delta$ combinations when sneakers are absent (*a*) or present (*b*). The data at the same pixel position across heatmaps in the same panel were generated from the same simulation run (see electronic supplementary material, figures S16–S19 for samples of evolutionary trajectories corresponding to four different pixels). Frequency of EPO stands for the proportion of EPO among all offspring produced at birth. Each simulation was run for 20 000 generations and the value at each pixel is the mean of the last 2000 generations. In each simulation, the population size was 5000, female fecundity $R$ was set to 100, the initial $h$ and $u$ values were both set to 0.5 and the initial frequency of the $a$ allele was set to 0.05 in the right column. (Online version in colour.)

**Figure 4.** Evolutionary cycles of the frequency of sneakers and the degrees of male help and female fidelity, when the sneakers have a competitive advantage in extra-pair fertilizations. The simulation parameters are $r = 1$, $\delta = 0$, $\beta = 2$, population size 10 000. We used a relatively high mutation rate of 0.05 at each locus and a mutation size distribution of zero mean and standard variation 0.05 to introduce more variation and speed up the simulation. See electronic supplementary material, figures S13 and S14 for more examples of evolutionary trajectories at different $\beta$ values. (Online version in colour.)

## 4. Discussion

Using a population genetics model and individual-based simulations, we studied the coevolutionary dynamics of female fidelity and male help in populations with two male ARTs (i.e. sneakers and bourgeois), and with EPO having higher or lower survival rates relative to WPO. We focused on analysing the conflicts between three parties: the females, the sneaker males and the bourgeois males. Female fidelity translates the competition between the two male ARTs into a zero-sum game within a generation, while over generations, the level of female fidelity coevolves with the level of male help and the relative frequencies of the two male ARTs.

First, we found that under a balanced sex ratio throughout the life cycle, the population growth rate is the highest when females are fully loyal and males spend all their time in brood care, even when the EPO had slightly higher survival rates than the WPO. However, sexual conflict and intrasexual competition can drive both female fidelity and male help to lower levels, and giving sneakers the opportunity to invade. This causes some females to remain unpaired and thus unable to reproduce, and the paired females suffer from high offspring mortality due to poor paternal care from their social mates. This result illustrates that sexual conflict and the competition between different types of males can lead to a 'tragedy of the commons', in concordance with empirical [47,48] and theoretical work [12,49]. A recent review [50] proposed the overarching term 'intraspecific adaptation load', which captures the competition and conflicts in our model and more general causes of conflicts with conspecifics (e.g. due to kin selection) at the expense of population fitness.

Second, our simulations revealed a novel result: when female fidelity is low (either fixed to low values or due to coevolution when EPO have a survival advantage) and the efficiency of mate guarding is intermediate (40%~80%), bourgeois males can evolve to providing extensive care by spending more than 90% of their time in brood care. At the same time, they try to guard their females as much as possible, despite high levels of cuckoldry (around 40% of offspring are fertilized by sneakers). In contrast to condition-dependent determination of male ARTs, where sneakers are hypothesized to do the 'best of a bad job' [51], we show that when the tactics are genetically determined, the bourgeois may instead be forced into a 'bad job' by providing extensive care while being heavily cuckolded as they cannot efficiently guard

their mates. This result corresponds nicely to empirical findings in the cichlid fish *Variabilichromis moori* [7] and may help explain the surprisingly low paternity in species with biparental care across taxa [8–11,52].

Mate guarding, as an 'ability of a portion of the population to control the access of others to potential mates' [53], plays an important role in the evolutionary dynamics driven by sexual conflict and the competition between sneakers and bourgeois males. Previous theoretical models often relied on assumptions that exaggerate the efficiency of male control, e.g. as long as a male attempts to guard his female, she has no opportunity to cheat [19,54–57]. Empirical studies demonstrated, however, mate guarding can often be inefficient due to various reasons, including: (1) female birds and mammals can often escape male paternity guarding, such as in the bluethroats (*Luscinia svecica*) [58], the yellow-breasted chats (*Icteria virens*) [59], the superb fairy-wrens (*Malurus cyaneus*) [60] and the Sika deer (*Cervus nippon*) [61]; (2) paired males may face a tradeoff between guarding and parental care, such as in black coucals (*Centropus grillii*), where parental care is provided by the males only, and once males start to incubate a (still incomplete) brood, they cannot prevent female EPC as efficiently as before, and consequently, EPO occur more often in the later-laid eggs [62]; and (3) females may use stored sperm of previous mates, which is often found in insects including burying beetles [17], golden egg bugs (*Phyllomorpha laciniata*) [63], and a bee species (*Ceratina nigrolabiata*) [64]. Our model shows that the bourgeois males can evolve to provide extensive parental care despite high levels of cuckoldry under imperfect mate guarding. This highlights the need of more theoretical work to bridge the gap between the common assumption of perfect mate guarding in existing models and the widespread inefficient mate guarding in nature.

Finally, we found that sexual conflict and the competition between male ARTs can drive the emergence of evolutionary cycles when sneakers are more competitive at extra-pair fertilization. Empirical evidence for adaptations to the sneaking tactic is abundant, such as smaller sizes [65], larger testes to body weight ratios and higher sperm densities in the ejaculates [66], as well as higher quality sperm [46]. We did not, however, find any empirical study documenting the evolutionary cycles predicted by our model, except some indirect, yet tantalizing support. Rios-Cardenas *et al.* [67] found negative frequency-dependent selection on the sneaker and bourgeois tactics in the swordtail fish (*Xiphophorus multilineatus*) using both mesocosm and field experiments, suggesting equal fitness

of the two tactics in evolutionary time scales. But the two tactics do not have equal fitness in their samples and vary in frequency in different years, including a sample with an extremely high proportion (86%) of sneakers. The authors therefore speculated that the frequency of sneakers might be fluctuating in a stable limit cycle [67]. Based on our theoretical prediction and the rare but suggestive empirical observations, we encourage more long-term field observations in populations with ARTs.

Our models showed as a proof-of-principle that the coevolution of female fidelity and male help driven by the sexual conflict and male ARTs can produce interesting evolutionary dynamics. A valuable future extension of our model will be to include variation in individual quality. With this, we could study condition-dependent reproductive tactics, the roles of assortative mating between fecund females and attractive males, and the effect of coercive mating by unwanted (probably low quality) males. In addition, individuals can also vary in their reproductive status in species of breeding asynchrony. In this case, a male might adopt the 'bourgeois' tactic while taking care of a brood and then switches to the 'sneaker' tactic afterwards. Such temporally flexible male ARTs affect the operational sex ratio of the population, which then feeds back to the benefit and opportunity cost of adopting each alternative tactic. It would be interesting to extend our model to include such individual variation as well. With this work, we showed the unexpected effects of imperfect mate guarding and male ARTs on the coevolutionary dynamics of female fidelity and male help, and we encourage future empirical tests of our predictions in species with these features.

Data accessibility. The MATLAB code corresponding to the analytical model and Python codes corresponding to the simulation model are provided in the GitHub repository: https://github.com/XiangyiLi/Coevolution-of-female-fidelity-and-male-help-in-populations-with-alternative-reproductive-tactics

Authors' contributions. X.-Y.L. and W.G. conceived the project; X.-Y.L. and A.M. designed the models and produced the results; X.-Y.L. wrote the paper with inputs from A.M. and W.G.

Competing interests. We declare we have no competing interests.

Acknowledgement. We thank Jussi Lehtonen (University of Sydney) and Yasuo Ihara (University of Tokyo) for their comments on an early version of the work, Simran Sandhu (University of Leicester) for helping to construct figure 1, and the three anonymous reviewers for their detailed and constructive suggestions. We thank the Swiss National Science Foundation and the Max Planck Society for funding.

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
