## [Reviewer comments · Proceedings of the Royal Society B: Biological Sciences]

Review History

RSPB-2019-2181.R0 (Original submission)

Review form: Reviewer 1

Recommendation

Major revision is needed (please make suggestions in comments)

Scientific importance: Is the manuscript an original and important contribution to its field?

Good

General interest: Is the paper of sufficient general interest?

Excellent

Quality of the paper: Is the overall quality of the paper suitable?

Good

Is the length of the paper justified?

Yes

Should the paper be seen by a specialist statistical reviewer?

No

Do you have any concerns about statistical analyses in this paper? If so, please specify them explicitly in your report.

No

It is a condition of publication that authors make their supporting data, code and materials available - either as supplementary material or hosted in an external repository. Please rate, if applicable, the supporting data on the following criteria.

Is it accessible?

Yes

Is it clear?

Yes

Is it adequate?

Yes

Do you have any ethical concerns with this paper?

No

Comments to the Author

This manuscript presents results of an individual-based simulation model that attempts to determine how intralocus sexual conflict affects the evolution of extra-pair copulation, mediated by female fidelity and male mate guarding. The main novel result is that intralocus sexual conflict can serve to promote female fidelity and male helping behaviour since extra-pair copulation with a high-quality male results in a cost in the quality of daughters.

This is a highly interesting result, but I'm a bit concerned that the model is overly complicated. I'm not an expert on EPCs, but have no other models incorporated costs to the female? The introduction gives the impression that this is the case. Since intralocus sexual conflict is essentially a cost to the female that increases with increasing quality of the extra-pair mate, couldn't this be incorporated into an analytical framework in a relatively simple way? It's not clear to me why such a complicated model is really necessary, and it makes it difficult to compare with previous work since it includes so many different parameters, all of which could partially contribute to the observed effect. I really encourage the authors to consider building a simple analytical model that captures the main features of the simulations, to form a conceptual bridge between the simulation model and previous analytical models.

Related to the point made above, must the cost come from intralocus sexual conflict? It seems to me that other scenarios are possible which make extra-pair matings with a highly attractive male more costly. For example if he's so much in demand that the female is forced to take any opportunity to mate with him, resulting in increased predation risk. I'm sure other plausible mechanisms are possible. I think it would be interesting to investigate the effect of a general such cost not related to sexual conflict, since this would increase the generality of the result.

Finally, are species that engage in duets and pas-de-deux usually highly sexually dimorphic? If not, then it's questionable how likely it is that intralocus sexual conflict constitutes a major cost. I think this needs to be mentioned in the discussion. All the more reason to consider a more general effect of the cost of EPCs scaling with male attractiveness, as well.

Minor comments:

Lines 70-75: You should cite Pischedda & Chippindale 2006 here, since they directly tested this hypothesis. <https://journals.plos.org/plosbiology/article?id=10.1371/journal.pbio.0040356>

The figure legends to Figures 3 and 4 are pretty brief. Especially Figure 4 needs more explanation in the figure text so that the reader can navigate it without having to read the main text.

I definitely don't think that the model needs to be made more complicated, but in nature female fidelity is probably quite context-dependent. It would be nice to include a short discussion of how this might affect the outcome.

Review form: Reviewer 2

Recommendation

Major revision is needed (please make suggestions in comments)

Scientific importance: Is the manuscript an original and important contribution to its field?

Good

General interest: Is the paper of sufficient general interest?

Good

Quality of the paper: Is the overall quality of the paper suitable?

Marginal

Is the length of the paper justified?

Yes

Should the paper be seen by a specialist statistical reviewer?

No

Do you have any concerns about statistical analyses in this paper? If so, please specify them explicitly in your report.

No

It is a condition of publication that authors make their supporting data, code and materials available - either as supplementary material or hosted in an external repository. Please rate, if applicable, the supporting data on the following criteria.

Is it accessible?

Yes

Is it clear?

Yes

Is it adequate?

Yes

Do you have any ethical concerns with this paper?

No

Comments to the Author

This study uses a mathematical model to examine the evolution of female fidelity and male parental help when there is a sexually-antagonistic trait. The main conclusion drawn from the model is that, when the sex ratio is male biased, high male helping and high female fidelity can evolve without efficient male mate guarding (L9-11, L244-248, L370-374). This cooperative outcome is thought to result from the inclusion of sexual-antagonism in the model (L214-216,

L251-252, L314-319) and the authors encourage empirical consideration of intralocus sexual conflict in systems with high female fidelity (L274-276, L320-321, L374-377).

Given that intralocus sexual conflict may result from sexual selection on one sex and opposing natural selection in both sexes (L67-75), the interaction between sexual conflict and sexual selection (in this case, mate choice through EPC) is a sensible research question. A further strength of the manuscript is the incorporation of relevant biological examples (e.g., L21-34, L72-75, L132, L254-260, L283-294, L301). However, I think that the model could be analysed more comprehensively, and I give some suggestions below.

Most importantly, I recommend that the main conclusions about the importance of sexual antagonism are interrogated explicitly. I recommend re-running the model without the I_s locus to directly compare the inclusion/exclusion of intralocus sexual conflict. This is necessary to justify the conclusion that intralocus sexual conflict explains the evolution of female fidelity under low mate guarding efficiency (e.g., L209). The authors state that females are selected to produce high-fecundity daughters, which means they favour mating with males of low quality and thus choose to mate with their partner rather than engaging in EPC. However, I believe that, for these parameters (male biased sex ratios) most EPC will come from unpaired males of low quality because helping (h) evolves to be high (paired males are therefore not available for EPC, eq on L118) and pairing occurs according to quality such that unpaired males are likely to be low quality (eq on L105). Thus, it seems reasonable to hypothesise that females evolve high fidelity to avoid EPC with unpaired males of low quality, running counter the explanation in the text. In general, I would also recommend unpacking exactly how intralocus sexual conflict is hypothesised to promote female fecundity because I found it a little unclear. In particular, I wasn't sure when/why females should be selected to produce high-quality daughters (and thus prefer a low-quality male) and not high-quality sons (e.g., L75-80, L209-210, L251-252, L316-317).

Male-biased sex ratios are key to the results, but the causes and likelihood of sex ratio bias are not discussed. This is crucial in establishing the relevance of the high female fidelity scenario that is outlined. In addition to a discussion, the authors could consider explicitly modelling how male-biased adult sex ratios could arise. For example, the probability of survival to reproductive age could depend on individual quality (not just parental quality and help).

I found the truncation of the Figure axes to be unsatisfying, particularly because the SI does not present the full results for regions where $\delta < 0.2$. I think that only one example is shown (Figure S3) so that it's difficult to get a sense of the overall dynamics when mate guarding is inefficient. I would imagine that $\delta = 0$ (males have no control over their partner's EPC) is a particularly important case and it is discussed on L246 and L372. To overcome the specific issue of population extinction (L180-184), perhaps the survival equation on L127 could be adjusted or, at least, justified? For example, offspring survival could be an additive function of parental quality, $s = (\alpha_f + h \cdot \alpha_m) / 2$. Changing this equation would also allow unpaired females to produce viable offspring, which might be interesting to investigate (I think this suggestion is slightly different from the issues discussed on L332-342).

Minor Comments:

L217-242, Figure 4, L271-272: I would suggest reducing the space dedicated to describing the case where female fidelity is subject to drift. Three paragraphs and one figure seems excessive to describe highly efficient mate guarding that prevents the female fidelity trait from being expressed. I would argue that this isn't as biologically interesting as cases where mate guarding is highly inefficient ($\delta = 0$).

L91: I think it'd be worth considering a model where A is fixed to consider species where the frequency of sneaker males can't readily evolve. In particular, I'd be interested to know whether sneakers are necessary for the cyclical dynamics under low mate guarding efficiency that are

presented in the supplementary information.

L357-358: Although it'd be difficult to examine many different trade-offs between male help and opportunities for EPC, maybe you could examine the case where there is no trade-off? That is, where male helping does not diminish their ability to mate with another female (remove h_i from ϕ_i on L118). This would allow you to examine the evolution of female fidelity when females can always choose from all males through EPC. This may be a good test for the hypothesis that female fidelity evolves because it's beneficial to remain mated with a low-quality male partner rather than seeking high-quality males through EPC.

L75-80: Given the previous result that females may prefer low quality males when selection is stronger for males than females, it seems strange that it is then assumed throughout that male competition for mating partners is stronger than female competition (Beta=2, L107). Perhaps the relative strength of selection on males/females is a key factor that determines whether females pursue EPC, which could be investigated by varying Beta?

Figs. 2 and 3: It would be easier to interpret if 'efficiency of mate guarding' was used for the x-axis on both figures rather than switching axes. I would also recommend organising the panels in the same order (rows with 'female fidelity' and 'sneakers' are switched between the two figures).

L133-135, Fig. 1: why are the l_m and l_f loci included? I expect them to simply fix for the beneficial allele and then be maintained at mutation-selection balance.

Fig. 3: Could you clarify that 'sneakers' are counted as 'unpaired males', if correct? It might also be worth clarifying that 'male help' is 'male help and mate guarding'. The 'median female quality' panel suggests that, in the region where there are sneaker males, the l_s locus reaches intermediate frequency. This might be an interesting case to describe in more detail.

Figs. S1 and S2: replace 'playboys' with 'sneakers'.

The deposition of all scripts should also be commended.

Review form: Reviewer 3

Recommendation

Major revision is needed (please make suggestions in comments)

Scientific importance: Is the manuscript an original and important contribution to its field?

Acceptable

General interest: Is the paper of sufficient general interest?

Good

Quality of the paper: Is the overall quality of the paper suitable?

Good

Is the length of the paper justified?

Yes

Should the paper be seen by a specialist statistical reviewer?

No

Do you have any concerns about statistical analyses in this paper? If so, please specify them explicitly in your report.

No

It is a condition of publication that authors make their supporting data, code and materials available - either as supplementary material or hosted in an external repository. Please rate, if applicable, the supporting data on the following criteria.

Is it accessible?

Yes

Is it clear?

Yes

Is it adequate?

Yes

Do you have any ethical concerns with this paper?

No

Comments to the Author

This is an interesting analysis on the evolution of female fidelity and male help. The authors use individual-based simulations to examine the impact of several parameters, including sex ratio biases and intra-locus sexual conflict, on the evolution of these two factors. This will be a useful addition to the literature, although I have two main concerns with the paper.

My first main concern with the paper deals with its use of biased sex ratios. The primary conclusion of this paper is that, under scenarios with a male-biased biased sex ratio, female fidelity may evolve because females prefer to stay with their current 'low-quality' mate, who will presumably produce high quality daughters due to sexual antagonism. The authors refer in the abstract to biased adult sex ratios. However, the model seems to be implementing a biased primary sex ratio – sex ratio at conception (lines 144-145 , 209-212). This is a very different scenario.

Fisher's well-established sex ratio principle states that sex ratios are approximately 1:1 at the end of reproductive investment due to the fact that every zygote has one mother and one father. If the birth of, for example, females is less common than males, individuals who produce more females gain a disproportionate advantage, as females have a higher reproductive value relative to males (Fisher 1930 for original argument, Hamilton 1967 "Extraordinary sex ratios" or Bull and Charnov 1988 "How fundamental are Fisherian sex ratios?" section 2.2.1 for a review of the argument).

Under the assumption of equal primary sex ratios, females equally gain equally from male and female offspring. Under a biased primary sex ratio, females are transmitting more through sons, although daughters have a higher reproductive value than sons in the next generation. While the authors suggest that the evolution of the focal behaviours is due to dynamics caused by the biased adult sex ratio, I am concerned that it is in fact due to the distortion of the primary sex ratio, and the biased transmission of female genes to her offspring.

This is critical to clarify as biases in the primary sex ratios are extremely rare in systems with genetic sex determination and, under Fisherian sex ratio selection, expected to be selected against. This holds true in birds [Liker et al. 2013 "The evolution of sex roles in birds is related to adult sex ratio"]. Highly biased adult sex ratios, however, are very common in many taxa, including birds [Pipoly et al. 2015 "The genetic sex-determination system predicts adult sex ratios in tetrapods"]. Clarifying this point has a direct impact on how broadly applicable the authors' results are. Ideally, the model should be implemented with a 50/50 primary sex ratio, and then random

culling can be performed to establish the adult sex ratio, which would test whether these effects are driven by the distortion of the primary sex ratio.

Generally, throughout the manuscript, the authors should be explicit about what type of sex ratio is being referred to and clarify the logic underlying their argument (In their discussion of female sex ratio adjustment, for example, the authors are referring to primary sex ratios (lines 261-262), but were just previously referring to adult sex ratios (line 247). Lines 314-317 are also based on females gaining a disproportionate fitness advantage through daughters rather than sons, and therefore on biases in the primary sex ratio. An enhanced discussion of the theory and empirical evidence for sex-ratio biases (how common are male-biased sex ratios in birds) would also be a helpful addition.

My second main concern with the paper is the assumption that intra-locus sexual conflict is driving the observed results. This argument needs to be laid out logically, and perhaps with better justification through simulations (if there was no intra-locus sexual conflict, i.e. $a_m = a_f = 1$, would the same results be observed?). This is connected to my concerns with the sex ratio, as if female transmission is occurring mainly through males due to the biased primary sex ratio, why should they have a “purely selfish motivation (to optimize their own fitness by producing high-quality daughters”? The authors should establish this logic more clearly, as it is the central claim of their paper.

Minor comments:

Line 37: “meta-trait” - could this be defined here?

Line 78: This should be made much more clear. Females may prefer low-quality males (who will presumably produce high-quality daughters) when selection on females is high. Could the hypothesis connecting this fact to female fidelity and/or male help be explicitly stated? It’s not clear whether, for instance, females mated with high quality males will prefer EPCs under these assumptions. The connection of male help to this model is also not clear.

Lines 126-127: I think this means that the female puts in full care into the brood regardless, and the male’s amount of care is modulated by the factor h . A sentence stating that in this line to go with the expression would be helpful.

Lines 133-134, Figure 1: I assume these are all autosomal loci, but it would be worthwhile to state this somewhere.

Figure 1c: This was very helpful.

Lines 136-139: I wonder if the authors could justify using three loci, only one of which is expressed in both sexes. This roughly corresponds to the influence of sex-specific expression on a particular trait that is nevertheless expressed in both sexes, but it would be nice to see an intuitive explanation along with the mathematical description.

Line 143: “ se ” should be “ set ”. I assume the extremely high mutation rate is simply to generate enough variation for these models, but perhaps it would be useful to mention here.

Lines 147-149: It would be nice to have a description of brood mortality in relation to male help here as well.

Lines 206-216 - If female transmission is mainly occurring through male offspring, why are females trying to maximize the fitness of daughters? This argument should be much more clear, although see the comments on sex ratio effects above.

Lines 212-214: This is a confusing based on the description of how mate pairing works in this species – it was stated earlier that high quality males are more likely to pair with high quality females. Should those females then preferentially find extra-pair matings, to avoid their high-quality social mate? Given that this is the fundamental claim of the paper, it is vital that this logic is clarified and a more detailed description provided.

Lines 251-252: I would like to see more justification that it is the intra-locus sexual conflict causing this result, as in lines 206-216. A better explanation of the logic underlying this claim is needed.

Lines 281-282: Could the authors clarify what is meant by frequent outliers here? Frequent instances of considerable male parental care?

Python code: I suggest that the authors change the term “playboy” to “sneaker” to better match the descriptions in the paper, at least in the comments on the code. However, this is a very minor point.

Figure S3: This is a nice illustration of the effects of finite population dynamics, as these cycles would not occur in simulations of an infinite population. Perhaps this point could be made in the main text when drift of female fidelity is discussed.

Decision letter (RSPB-2019-2181.R0)

01-Nov-2019

Dear Dr Li,

I am writing to inform you that your manuscript RSPB-2019-2181 entitled "Coevolution of female fidelity and male help under intralocus sexual conflict" has, in its current form, been rejected for publication in Proceedings B.

This action has been taken on the advice of referees, who have recommended that substantial revisions are necessary. With this in mind we would be happy to consider a resubmission, provided the comments of the referees are fully addressed. However please note that this is not a provisional acceptance.

To upload a resubmitted manuscript, log into <http://mc.manuscriptcentral.com/prsb> and enter your Author Centre, where you will find your manuscript title listed under "Manuscripts with

Decisions." Under "Actions," click on "Create a Resubmission." Please be sure to indicate in your cover letter that it is a resubmission, and supply the previous reference number.

Sincerely,
Professor Loeske Kruuk
mailto: proceedingsb@royalsociety.org

Associate Editor

Board Member: 1

Comments to Author:

In this paper, Li and Goymann use individual-based simulations to address the factors that explain variation across species in levels of female fidelity and extra pair paternity, presenting a model that investigates effects of intralocus sexual conflict on these parameters. The model presented is interesting, and the paper was therefore sent for review to three referees, who have each provided very constructive feedback and extensive comments. These reviews all concur that the paper requires major revision.

Referee 1 believes the model to be overly complicated, asks for clarification on whether other models have previously modelled costs to females, and also recommended the authors seek to simplify within an analytical framework. They also ask whether costs that are unrelated to intralocus conflict can be included that increase the generality of the result.

Referee 2 concludes that the overall significance of the work is difficult to judge since not enough work has gone into validating the role of sexual antagonism; more model analysis is required to justify the interpretation that intralocus conflict drives the evolution of female fidelity. The referee provides valuable suggestions on how to proceed. The referee also states more results are needed to be shown for the parameter space where male mate guarding is insufficient. The referee provides alternative credible interpretations, and questions the assumption of why females should be selected to produce high quality daughters. They suggest the authors also model how male biased sex ratios could arise.

Referee 3 has two main concerns requiring addressing; firstly around the model implementing a biased primary sex ratio rather than adult sex ratio, and secondly the assumption that intralocus conflict is driving the results.

All referee comments will require careful consideration and incorporation into a revised manuscript.

Reviewer(s)' Comments to Author:

Referee: 1

Comments to the Author(s)

This manuscript presents results of an individual-based simulation model that attempts to determine how intralocus sexual conflict affects the evolution of extra-pair copulation, mediated by female fidelity and male mate guarding. The main novel result is that intralocus sexual conflict can serve to promote female fidelity and male helping behaviour since extra-pair copulation with a high-quality male results in a cost in the quality of daughters.

This is a highly interesting result, but I'm a bit concerned that the model is overly complicated. I'm not an expert on EPCs, but have no other models incorporated costs to the female? The introduction gives the impression that this is the case. Since intralocus sexual conflict is essentially a cost to the female that increases with increasing quality of the extra-pair mate, couldn't this be incorporated into an analytical framework in a relatively simple way? It's not clear to me why such a complicated model is really necessary, and it makes it difficult to compare with previous work since it includes so many different parameters, all of which could partially contribute to the observed effect. I really encourage the authors to consider building a simple

analytical model that captures the main features of the simulations, to form a conceptual bridge between the simulation model and previous analytical models.

Related to the point made above, must the cost come from intralocus sexual conflict? It seems to me that other scenarios are possible which make extra-pair matings with a highly attractive male more costly. For example if he's so much in demand that the female is forced to take any opportunity to mate with him, resulting in increased predation risk. I'm sure other plausible mechanisms are possible. I think it would be interesting to investigate the effect of a general such cost not related to sexual conflict, since this would increase the generality of the result.

Finally, are species that engage in duets and pas-de-deux usually highly sexually dimorphic? If not, then it's questionable how likely it is that intralocus sexual conflict constitutes a major cost. I think this needs to be mentioned in the discussion. All the more reason to consider a more general effect of the cost of EPCs scaling with male attractiveness, as well.

Minor comments:

Lines 70-75: You should cite Pischedda & Chippindale 2006 here, since they directly tested this hypothesis. <https://journals.plos.org/plosbiology/article?id=10.1371/journal.pbio.0040356>

The figure legends to Figures 3 and 4 are pretty brief. Especially Figure 4 needs more explanation in the figure text so that the reader can navigate it without having to read the main text.

I definitely don't think that the model needs to be made more complicated, but in nature female fidelity is probably quite context-dependent. It would be nice to include a short discussion of how this might affect the outcome.

Referee: 2

Comments to the Author(s)

This study uses a mathematical model to examine the evolution of female fidelity and male parental help when there is a sexually-antagonistic trait. The main conclusion drawn from the model is that, when the sex ratio is male biased, high male helping and high female fidelity can evolve without efficient male mate guarding (L9-11, L244-248, L370-374). This cooperative outcome is thought to result from the inclusion of sexual-antagonism in the model (L214-216, L251-252, L314-319) and the authors encourage empirical consideration of intralocus sexual conflict in systems with high female fidelity (L274-276, L320-321, L374-377).

Given that intralocus sexual conflict may result from sexual selection on one sex and opposing natural selection in both sexes (L67-75), the interaction between sexual conflict and sexual selection (in this case, mate choice through EPC) is a sensible research question. A further strength of the manuscript is the incorporation of relevant biological examples (e.g., L21-34, L72-75, L132, L254-260, L283-294, L301). However, I think that the model could be analysed more comprehensively, and I give some suggestions below.

Most importantly, I recommend that the main conclusions about the importance of sexual antagonism are interrogated explicitly. I recommend re-running the model without the I_s locus to directly compare the inclusion/exclusion of intralocus sexual conflict. This is necessary to justify the conclusion that intralocus sexual conflict explains the evolution of female fidelity under low mate guarding efficiency (e.g., L209). The authors state that females are selected to produce high-fecundity daughters, which means they favour mating with males of low quality and thus choose to mate with their partner rather than engaging in EPC. However, I believe that, for these parameters (male biased sex ratios) most EPC will come from unpaired males of low quality because helping (h) evolves to be high (paired males are therefore not available for EPC, eq on L118) and pairing occurs according to quality such that unpaired males are likely to be low quality (eq on L105). Thus, it seems reasonable to hypothesise that females evolve high fidelity to avoid EPC with unpaired males of low quality, running counter the explanation in the text. In

general, I would also recommend unpacking exactly how intralocus sexual conflict is hypothesised to promote female fecundity because I found it a little unclear. In particular, I wasn't sure when/why females should be selected to produce high-quality daughters (and thus prefer a low-quality male) and not high-quality sons (e.g., L75-80, L209-210, L251-252, L316-317).

Male-biased sex ratios are key to the results, but the causes and likelihood of sex ratio bias are not discussed. This is crucial in establishing the relevance of the high female fidelity scenario that is outlined. In addition to a discussion, the authors could consider explicitly modelling how male-biased adult sex ratios could arise. For example, the probability of survival to reproductive age could depend on individual quality (not just parental quality and help).

I found the truncation of the Figure axes to be unsatisfying, particularly because the SI does not present the full results for regions where $\delta < 0.2$. I think that only one example is shown (Figure S3) so that it's difficult to get a sense of the overall dynamics when mate guarding is inefficient. I would imagine that $\delta = 0$ (males have no control over their partner's EPC) is a particularly important case and it is discussed on L246 and L372. To overcome the specific issue of population extinction (L180-184), perhaps the survival equation on L127 could be adjusted or, at least, justified? For example, offspring survival could be an additive function of parental quality, $s = (\alpha_f + h \cdot \alpha_m) / 2$. Changing this equation would also allow unpaired females to produce viable offspring, which might be interesting to investigate (I think this suggestion is slightly different from the issues discussed on L332-342).

Minor Comments:

L217-242, Figure 4, L271-272: I would suggest reducing the space dedicated to describing the case where female fidelity is subject to drift. Three paragraphs and one figure seems excessive to describe highly efficient mate guarding that prevents the female fidelity trait from being expressed. I would argue that this isn't as biologically interesting as cases where mate guarding is highly inefficient ($\delta = 0$).

L91: I think it'd be worth considering a model where A is fixed to consider species where the frequency of sneaker males can't readily evolve. In particular, I'd be interested to know whether sneakers are necessary for the cyclical dynamics under low mate guarding efficiency that are presented in the supplementary information.

L357-358: Although it'd be difficult to examine many different trade-offs between male help and opportunities for EPC, maybe you could examine the case where there is no trade-off? That is, where male helping does not diminish their ability to mate with another female (remove h_i from ϕ_i on L118). This would allow you to examine the evolution of female fidelity when females can always choose from all males through EPC. This may be a good test for the hypothesis that female fidelity evolves because it's beneficial to remain mated with a low-quality male partner rather than seeking high-quality males through EPC.

L75-80: Given the previous result that females may prefer low quality males when selection is stronger for males than females, it seems strange that it is then assumed throughout that male competition for mating partners is stronger than female competition ($\beta = 2$, L107). Perhaps the relative strength of selection on males/females is a key factor that determines whether females pursue EPC, which could be investigated by varying β ?

Figs. 2 and 3: It would be easier to interpret if 'efficiency of mate guarding' was used for the x-axis on both figures rather than switching axes. I would also recommend organising the panels in the same order (rows with 'female fidelity' and 'sneakers' are switched between the two figures).

L133-135, Fig. 1: why are the l_m and l_f loci included? I expect them to simply fix for the beneficial allele and then be maintained at mutation-selection balance.

Fig. 3: Could you clarify that 'sneakers' are counted as 'unpaired males', if correct? It might also be worth clarifying that 'male help' is 'male help and mate guarding'. The 'median female quality' panel suggests that, in the region where there are sneaker males, the L_s locus reaches intermediate frequency. This might be an interesting case to describe in more detail.

Figs. S1 and S2: replace 'playboys' with 'sneakers'.

The deposition of all scripts should also be commended.

Referee: 3

Comments to the Author(s)

This is an interesting analysis on the evolution of female fidelity and male help. The authors use individual-based simulations to examine the impact of several parameters, including sex ratio biases and intra-locus sexual conflict, on the evolution of these two factors. This will be a useful addition to the literature, although I have two main concerns with the paper.

My first main concern with the paper deals with its use of biased sex ratios. The primary conclusion of this paper is that, under scenarios with a male-biased biased sex ratio, female fidelity may evolve because females prefer to stay with their current 'low-quality' mate, who will presumably produce high quality daughters due to sexual antagonism. The authors refer in the abstract to biased adult sex ratios. However, the model seems to be implementing a biased primary sex ratio – sex ratio at conception (lines 144-145 , 209-212). This is a very different scenario.

Fisher's well-established sex ratio principle states that sex ratios are approximately 1:1 at the end of reproductive investment due to the fact that every zygote has one mother and one father. If the birth of, for example, females is less common than males, individuals who produce more females gain a disproportionate advantage, as females have a higher reproductive value relative to males (Fisher 1930 for original argument, Hamilton 1967 "Extraordinary sex ratios" or Bull and Charnov 1988 "How fundamental are Fisherian sex ratios?" section 2.2.1 for a review of the argument).

Under the assumption of equal primary sex ratios, females equally gain equally from male and female offspring. Under a biased primary sex ratio, females are transmitting more through sons, although daughters have a higher reproductive value than sons in the next generation. While the authors suggest that the evolution of the focal behaviours is due to dynamics caused by the biased adult sex ratio, I am concerned that it is in fact due to the distortion of the primary sex ratio, and the biased transmission of female genes to her offspring.

This is critical to clarify as biases in the primary sex ratios are extremely rare in systems with genetic sex determination and, under Fisherian sex ratio selection, expected to be selected against. This holds true in birds [Liker et al. 2013 "The evolution of sex roles in birds is related to adult sex ratio"]. Highly biased adult sex ratios, however, are very common in many taxa, including birds [Pipoly et al. 2015 "The genetic sex-determination system predicts adult sex ratios in tetrapods"]. Clarifying this point has a direct impact on how broadly applicable the authors' results are. Ideally, the model should be implemented with a 50/50 primary sex ratio, and then random culling can be performed to establish the adult sex ratio, which would test whether these effects are driven by the distortion of the primary sex ratio.

Generally, throughout the manuscript, the authors should be explicit about what type of sex ratio is being referred to and clarify the logic underlying their argument (In their discussion of female sex ratio adjustment, for example, the authors are referring to primary sex ratios (lines 261-262), but were just previously referring to adult sex ratios (line 247). Lines 314-317 are also based on females gaining a disproportionate fitness advantage through daughters rather than sons, and

therefore on biases in the primary sex ratio. An enhanced discussion of the theory and empirical evidence for sex-ratio biases (how common are male-biased sex ratios in birds) would also be a helpful addition.

My second main concern with the paper is the assumption that intra-locus sexual conflict is driving the observed results. This argument needs to be laid out logically, and perhaps with better justification through simulations (if there was no intra-locus sexual conflict, i.e. $a_m = a_f = 1$, would the same results be observed?). This is connected to my concerns with the sex ratio, as if female transmission is occurring mainly through males due to the biased primary sex ratio, why should they have a “purely selfish motivation (to optimize their own fitness by producing high-quality daughters”? The authors should establish this logic more clearly, as it is the central claim of their paper.

Minor comments:

Line 37: “meta-trait” - could this be defined here?

Line 78: This should be made much more clear. Females may prefer low-quality males (who will presumably produce high-quality daughters) when selection on females is high. Could the hypothesis connecting this fact to female fidelity and/or male help be explicitly stated? It’s not clear whether, for instance, females mated with high quality males will prefer EPCs under these assumptions. The connection of male help to this model is also not clear.

Lines 126-127: I think this means that the female puts in full care into the brood regardless, and the male’s amount of care is modulated by the factor h . A sentence stating that in this line to go with the expression would be helpful.

Lines 133-134, Figure 1: I assume these are all autosomal loci, but it would be worthwhile to state this somewhere.

Figure 1c: This was very helpful.

Lines 136-139: I wonder if the authors could justify using three loci, only one of which is expressed in both sexes. This roughly corresponds to the influence of sex-specific expression on a particular trait that is nevertheless expressed in both sexes, but it would be nice to see an intuitive explanation along with the mathematical description.

Line 143: “se” should be “set”. I assume the extremely high mutation rate is simply to generate enough variation for these models, but perhaps it would be useful to mention here.

Lines 147-149: It would be nice to have a description of brood mortality in relation to male help here as well.

Lines 206-216 - If female transmission is mainly occurring through male offspring, why are females trying to maximize the fitness of daughters? This argument should be much more clear, although see the comments on sex ratio effects above.

Lines 212-214: This is a confusing based on the description of how mate pairing works in this species - it was stated earlier that high quality males are more likely to pair with high quality females. Should those females then preferentially find extra-pair matings, to avoid their high-quality social mate? Given that this is the fundamental claim of the paper, it is vital that this logic is clarified and a more detailed description provided.

Lines 251-252: I would like to see more justification that it is the intra-locus sexual conflict causing this result, as in lines 206-216. A better explanation of the logic underlying this claim is needed.

Lines 281-282: Could the authors clarify what is meant by frequent outliers here? Frequent instances of considerable male parental care?

Python code: I suggest that the authors change the term “playboy” to “sneaker” to better match the descriptions in the paper, at least in the comments on the code. However, this is a very minor point.

Figure S3: This is a nice illustration of the effects of finite population dynamics, as these cycles would not occur in simulations of an infinite population. Perhaps this point could be made in the main text when drift of female fidelity is discussed.

Author's Response to Decision Letter for (RSPB-2019-2181.R0)

See Appendix A.

RSPB-2020-2371.R0

Review form: Reviewer 1

Recommendation

Major revision is needed (please make suggestions in comments)

Scientific importance: Is the manuscript an original and important contribution to its field?

Good

General interest: Is the paper of sufficient general interest?

Good

Quality of the paper: Is the overall quality of the paper suitable?

Good

Is the length of the paper justified?

Yes

Should the paper be seen by a specialist statistical reviewer?

No

Do you have any concerns about statistical analyses in this paper? If so, please specify them explicitly in your report.

No

It is a condition of publication that authors make their supporting data, code and materials available - either as supplementary material or hosted in an external repository. Please rate, if applicable, the supporting data on the following criteria.

Is it accessible?

Yes

Is it clear?

Yes

Is it adequate?

Yes

Do you have any ethical concerns with this paper?

No

Comments to the Author

This manuscript presents a model of the evolution of female fidelity and male parental care in species with alternative reproductive tactics. They find that for fixed values of female fidelity and male help, population fitness is highest at high levels of both fidelity and help. However when these traits are allowed to evolve, the outcome changes. Female fidelity only remains high if extra-pair offspring have reduced survival, and males only care if either extra-pair offspring have reduced survival OR the efficiency of mate guarding is high. Interestingly, results from the analytical model suggest that at intermediate rates of mate guarding efficiency, males will still care despite high rates of extra-pair offspring. Results from the simulation model suggest that evolutionary cycles in male care and female fidelity can also occur.

The model and results are somewhat complicated, but the results are timely and interesting. My main criticism is that the reader gets the impression that the model seems to be set-up particularly to explain observations in systems similar to the cichlid system mentioned in the introduction. Although other systems are mentioned briefly in the discussion on lines 400-409, I felt that the generality of the results could be made more explicit by:

1. Talking more about the implications of trade-offs between guarding and caring. If parental care mainly consists of e.g. incubating eggs while the female forages or obtaining food for offspring (as in birds), then the efficiency of mate guarding while caring will be very low. How do the model results align with empirical data from systems where efficiency of mate guarding is low?
2. Discussing implications with respect to sexual conflict. The model is set up such that females are either willing to mate with extra pair males or not, but in many species with alternative reproductive tactics the "sneaker" males engage in coercive mating. Is it possible to investigate this sort of scenario in your model, or would it require additional work? I'm thinking that when $r < 1$ then this could be similar to a coercive mating scenario, where mating with unpreferred coercive males has negative effects on offspring survival. I'm not sure whether the female fidelity locus can be interpreted in a sexual conflict context or not, though - maybe as females with underlying resistance traits to sneaky matings or not? At the very least, I think it would be worth bringing up whether these results are equally applicable to a sexual conflict context or not, and if so, then how.

Another less important point is that I was missing some sort of motivation why the simulations assumed that sneakers could have an exaggerated advantage, while the analytical models assumed a linear relationship. Why not incorporate this in the analytical component as well? I would have also liked some sort of prediction as to what the expected difference would be.

Minor comments

Line 23: change "with genetically determined two" to "with two genetically determined"

Line 98: remove "the" from "although the cooperation"

Line 115-116: Have you explored at all what happens when this assumption is relaxed? The manuscript already presents a lot of results, so I don't mean you need to do this and report it, I'm just curious.

Lines 179-183: Here I think it wouldn't hurt to be a bit more explicit, and explain that the ART locus evolves by changing frequencies while the fidelity and help loci evolve as quantitative traits.

Line 186: change "Without" to "Unless"

Lines 242-283: Is there any biological justification for these scenarios, or are they just carried out to better understand the dynamics when both can vary?

Line 271: change "evolve" to "evolves"

Line 272: change "is" to "are"

Line 302: change "can sustain" to "is sustainable"

Line 348: change "sneakers (now suffer" to "sneakers (who now suffer"

Line 361: I don't know if I agree that this model is particularly simple - there are very many parameters that can be varied.

Review form: Reviewer 2

Recommendation

Accept with minor revision (please list in comments)

Scientific importance: Is the manuscript an original and important contribution to its field?

Good

General interest: Is the paper of sufficient general interest?

Good

Quality of the paper: Is the overall quality of the paper suitable?

Acceptable

Is the length of the paper justified?

Yes

Should the paper be seen by a specialist statistical reviewer?

No

Do you have any concerns about statistical analyses in this paper? If so, please specify them explicitly in your report.

No

It is a condition of publication that authors make their supporting data, code and materials available - either as supplementary material or hosted in an external repository. Please rate, if applicable, the supporting data on the following criteria.

Is it accessible?

Yes

Is it clear?

Yes

Is it adequate?

Yes

Do you have any ethical concerns with this paper?

No

Comments to the Author

The previous version of this manuscript focussed on the interaction between sexual antagonism and mate choice. The revised version does not include sexually antagonistic selection and focusses instead on the coevolution of male help and female fidelity in the presence of alternative

male reproductive tactics (sneaker and bourgeois males). The key result is that males can evolve to provide a lot of parental care despite high percentages of the brood being sired by other males (specifically, by sneaker males). The authors link this result to observations of fish that care for nests containing many offspring of other males. The model simplification has made the presentation much clearer and I have only minor suggestions for improvement.

The introduction says: “Theoretical models have identified... inability of females to fully compensate the loss of male care... [could cause males to] be ‘blackmailed’ to continue providing care by their paired mates, because otherwise their own genetic offspring (the ‘hostages’) will also suffer” (L47-51). This seems to be the situation described in the model – females can’t compensate for loss of male care (only females paired with bourgeois males reproduce) and males are ‘blackmailed’ into providing help for extra-pair offspring along with their own. Given this apparent precedent, I would suggest clarifying the precise novelty of the result that males provide parental care to broods with high extra-pair paternity rates.

The model now includes a difference in survival between offspring produced from within-pair and extra-pair matings. This is important to most of the results so I would suggest that the authors add an explicit justification. I suppose that females choose high quality males when they seek out extra-pair matings and therefore have high fitness offspring. However, male quality has been removed from the model.

On a related note, it is important that the cyclical dynamics (Figure 3) occur when offspring from within-pair and from extra-pair matings have the same fitness ($r=1$). This means that female fidelity is not evolving due to direct fitness benefits to offspring. Instead, female (in)fidelity evolves to select males that will father sons that will be more successful, as occurs in classical sexual selection. Whether within-pair or extra-pair mates father more successful sons depends on the position within the cycle. The cycles occur when bourgeois males gain diminishing returns from EPC (L333). This means that bourgeois males are more effective when male help is low, which I think might cause cycling even without the inclusion of sneakers. I suggest the authors check whether cycles occur without sneakers.

In the population genetics model, I was expecting to see that the proportions of different genotypes were followed, rather than the absolute number of individuals. That is, I would remove R and Eq 5 and 6 and normalize M_i and F_j by the $\sum_i (M_i)$ and $\sum_j (F_j)$ so that they represent genotype frequencies (summing to 1). The aim of the model is to find the equilibrium frequency of sneaker males (L165). I expect that this adjustment might help to find this equilibrium frequency analytically.

The results where male help evolves to zero are superficially strange (top left of Figure 3). When $h=0$, all offspring die (L125) and both bourgeois and sneaker males spend all their time looking for mates. I gather that the population is only viable here because mutation-selection balance means $h \neq 0$. Given the new focus on population mean fitness (Figure 1) and the ‘tragedy of the commons’ (L100, L230, L291), perhaps you could point out that populations may evolve to extinction when this happens. E.g., by running some simulations with lower R. In general, I also think it would be useful to point out that sneakers are effectively the same as bourgeois males but with an allele that means they always have $h=0$ (until later in the paper where sneakers have specific adaptations that make them especially effective in EPC).

Review form: Reviewer 3

Recommendation

Accept with minor revision (please list in comments)

Scientific importance: Is the manuscript an original and important contribution to its field?
Good

General interest: Is the paper of sufficient general interest?
Good

Quality of the paper: Is the overall quality of the paper suitable?
Good

Is the length of the paper justified?
Yes

Should the paper be seen by a specialist statistical reviewer?
No

Do you have any concerns about statistical analyses in this paper? If so, please specify them explicitly in your report.
No

It is a condition of publication that authors make their supporting data, code and materials available - either as supplementary material or hosted in an external repository. Please rate, if applicable, the supporting data on the following criteria.

Is it accessible?
Yes

Is it clear?
Yes

Is it adequate?
Yes

Do you have any ethical concerns with this paper?
No

Comments to the Author

The manuscript presents a simplified version of the previously presented model, with the removal of the sexually antagonistic component. This makes the dynamics of the model, and the manuscript as a whole, much easier to follow. I have a few additional comments on the manuscript, stated below:

Equation 1 - There is a space missing after the comma

Line 148-149 - It would be nice if δ_2 could be defined in the same way as δ_1 , so it is clear that these are comparable numbers or not. I am not sure about the meaning of "increment" in this context, specifically. It might also be worthwhile to change the numbers 1 and 2 to something more informative, e.g. "wp" for within-pair and "ep" for between pair.

Lines 123 - 124 - Are there examples of either of these cases that could be cited here? The potential mechanisms underlying these effects is not obvious.

Line 186 - "without" should be "unless"

Lines 243-244 - It would be helpful to note explicitly where the results transition from the results of the analytical model of fitness landscapes to the individual simulation results. It is clear from the figures, but would be useful to have in the text as well.

Line 272 - "is" should be "are"

Line 278 - I'm not sure what "from the same realization of simulation" means here. The same parameter combinations?

Line 302 - "can sustain" could be "can be sustained" or "can sustain itself"

Line 304 - I would suggest that the authors are explicit about this being the peak of the population fitness landscape, both here and elsewhere in the manuscript where the fitness landscape is mentioned, to avoid confusion with a sex-specific fitness landscape.

Lines 337-351 - This pattern bears a superficial resemblance to a Fisherian "sexy-sons" mechanism - namely, the increase in fitness of a mother is through the increased fitness of her sons (here, through their strategy) and their daughters tend to mate with individual who share the sons phenotype. In a Fisherian model, the advantage to sons would be solely through their mating success due to some ornament, rather than the strategy fitness benefit here. It might be worthwhile to look into the parallels between the two models.

Line 358 - Variations should be variation.

Lines 378 - Female should be females

Lines 378 - 380 - This seems like a slightly different point, as sexually antagonistic alleles are expressed, by definition, in both sexes while here you are considering the evolution of a male-specific behaviour that reduces female fitness due to lack of parental care.

Line 416 - Sperms should be sperm

Line 438 - It is not clear why the balance of the trade-off should depend specifically on the adult sex ratio. Could this be elaborated upon slightly?

Decision letter (RSPB-2020-2371.R0)

09-Nov-2020

Dear Dr Li:

Your manuscript has now been peer reviewed and the reviews have been assessed by an Associate Editor. The reviewers' comments (not including confidential comments to the Editor) and the comments from the Associate Editor are included at the end of this email for your reference. As you will see, the reviewers and the Associate Editor are all agreed that the revised version of the paper is much clearer, but they have also raised some concerns with your manuscript. We therefore invite you to revise your manuscript to address these concerns.

The referees' comments will require some substantial revisions, so please contact us if these will not be feasible within the time-frame allocated (see below). We do not allow multiple rounds of revision so we urge you to make every effort to fully address all of the comments at this stage. Your manuscript will be sent back to one or more of the original reviewers for assessment (if the original reviewers are not available, we may invite new reviewers). Please note that we cannot guarantee eventual acceptance of your manuscript at this stage.

Research ethics:

Use of animals and field studies:

It is a condition of publication that you make available the data and research materials supporting the results in the article (<https://royalsociety.org/journals/authors/author-guidelines/#data>). Datasets should be deposited in an appropriate publicly available repository and details of the associated accession number, link or DOI to the datasets must be included in the Data Accessibility section of the article (<https://royalsociety.org/journals/ethics-policies/data-sharing-mining/>). Reference(s) to datasets should also be included in the reference list of the article with DOIs (where available).

All supplementary materials accompanying an accepted article will be treated as in their final form. They will be published alongside the paper on the journal website and posted on the online

figshare repository. Files on figshare will be made available approximately one week before the accompanying article so that the supplementary material can be attributed a unique DOI. Please try to submit all supplementary material as a single file.

Please submit a copy of your revised paper within three weeks. If we do not hear from you within this time your manuscript will be rejected. If you are unable to meet this deadline please let us know as soon as possible, as we may be able to grant a short extension.

Best wishes,
Professor Loeske Kruuk
mailto:proceedingsb@royalsociety.org

Associate Editor

Comments to Author:

The authors have performed a thorough revision to their models and associated paper; an observation noted by all three referees. The previous version of this paper focused on modelling the coevolution of female fidelity and male help under intralocus sexual conflict. Following advice from the referees the authors have simplified their model to focus specifically on the coevolution of male help and female fidelity in the presence of alternative male reproductive tactics. They show that when the efficiency of mate guarding is intermediate, males can evolve to specialise in providing high parental care despite high EPFs by sneaker males. The results are scaffolded on observations of cichlids where brood-tending males provide care to their brood despite losing high levels of paternity.

While the paper is very interesting, and the models and revision very thorough, the referees have expressed several insightful queries and made constructive suggestions for further analysis that will require a further revision prior to the study being suitable for publication in this journal. In terms of the suitability of this paper for Proceedings B, two points require attention of the authors:

Firstly, the models are now quite limited in scope, and the novelty of the new models and their conclusions is less clear – as pointed out by Referee 2. The authors refer to previous theoretical studies with similar conclusions (lines 46-53), and thus it is very important that the authors clarify the novelty of this result; since this will be key to determining suitability of this study in Proceedings B.

Related to this, the conclusions of the study are now more limited than previously; applicable to species where male care is required for reproduction and that also exhibit alternative male tactics. While the authors link this to some fish species exhibiting male brood care, the study would be improved (and better suited to the readership of Proceedings B) by further generalization to other systems, and Referee 1 has insightful suggestions here that require careful thought.

Reviewer(s)' Comments to Author:

Referee: 1

Comments to the Author(s).

This manuscript presents a model of the evolution of female fidelity and male parental care in species with alternative reproductive tactics. They find that for fixed values of female fidelity and

male help, population fitness is highest at high levels of both fidelity and help. However when these traits are allowed to evolve, the outcome changes. Female fidelity only remains high if extra-pair offspring have reduced survival, and males only care if either extra-pair offspring have reduced survival OR the efficiency of mate guarding is high. Interestingly, results from the analytical model suggest that at intermediate rates of mate guarding efficiency, males will still care despite high rates of extra-pair offspring. Results from the simulation model suggest that evolutionary cycles in male care and female fidelity can also occur.

The model and results are somewhat complicated, but the results are timely and interesting. My main criticism is that the reader gets the impression that the model seems to be set-up particularly to explain observations in systems similar to the cichlid system mentioned in the introduction. Although other systems are mentioned briefly in the discussion on lines 400-409, I felt that the generality of the results could be made more explicit by:

1. Talking more about the implications of trade-offs between guarding and caring. If parental care mainly consists of e.g. incubating eggs while the female forages or obtaining food for offspring (as in birds), then the efficiency of mate guarding while caring will be very low. How do the model results align with empirical data from systems where efficiency of mate guarding is low?

2. Discussing implications with respect to sexual conflict. The model is set up such that females are either willing to mate with extra pair males or not, but in many species with alternative reproductive tactics the "sneaker" males engage in coercive mating. Is it possible to investigate this sort of scenario in your model, or would it require additional work? I'm thinking that when $r < 1$ then this could be similar to a coercive mating scenario, where mating with unpreferred coercive males has negative effects on offspring survival. I'm not sure whether the female fidelity locus can be interpreted in a sexual conflict context or not, though - maybe as females with underlying resistance traits to sneaky matings or not? At the very least, I think it would be worth bringing up whether these results are equally applicable to a sexual conflict context or not, and if so, then how.

Another less important point is that I was missing some sort of motivation why the simulations assumed that sneakers could have an exaggerated advantage, while the analytical models assumed a linear relationship. Why not incorporate this in the analytical component as well? I would have also liked some sort of prediction as to what the expected difference would be.

Minor comments

Line 23: change "with genetically determined two" to "with two genetically determined"

Line 98: remove "the" from "although the cooperation"

Line 115-116: Have you explored at all what happens when this assumption is relaxed? The manuscript already presents a lot of results, so I don't mean you need to do this and report it, I'm just curious.

Lines 179-183: Here I think it wouldn't hurt to be a bit more explicit, and explain that the ART locus evolves by changing frequencies while the fidelity and help loci evolve as quantitative traits.

Line 186: change "Without" to "Unless"

Lines 242-283: Is there any biological justification for these scenarios, or are they just carried out to better understand the dynamics when both can vary?

Line 271: change "evolve" to "evolves"

Line 272: change "is" to "are"

Line 302: change "can sustain" to "is sustainable"

Line 348: change "sneakers (now suffer" to "sneakers (who now suffer"

Line 361: I don't know if I agree that this model is particularly simple - there are very many parameters that can be varied.

Referee: 2

Comments to the Author(s).

The previous version of this manuscript focussed on the interaction between sexual antagonism and mate choice. The revised version does not include sexually antagonistic selection and focusses instead on the coevolution of male help and female fidelity in the presence of alternative male reproductive tactics (sneaker and bourgeois males). The key result is that males can evolve to provide a lot of parental care despite high percentages of the brood being sired by other males (specifically, by sneaker males). The authors link this result to observations of fish that care for nests containing many offspring of other males. The model simplification has made the presentation much clearer and I have only minor suggestions for improvement.

The introduction says: "Theoretical models have identified... inability of females to fully compensate the loss of male care... [could cause males to] be 'blackmailed' to continue providing care by their paired mates, because otherwise their own genetic offspring (the 'hostages') will also suffer" (L47-51). This seems to be the situation described in the model - females can't compensate for loss of male care (only females paired with bourgeois males reproduce) and males are 'blackmailed' into providing help for extra-pair offspring along with their own. Given this apparent precedent, I would suggest clarifying the precise novelty of the result that males provide parental care to broods with high extra-pair paternity rates.

The model now includes a difference in survival between offspring produced from within-pair and extra-pair matings. This is important to most of the results so I would suggest that the authors add an explicit justification. I suppose that females choose high quality males when they seek out extra-pair matings and therefore have high fitness offspring. However, male quality has been removed from the model.

On a related note, it is important that the cyclical dynamics (Figure 3) occur when offspring from within-pair and from extra-pair matings have the same fitness ($r=1$). This means that female fidelity is not evolving due to direct fitness benefits to offspring. Instead, female (in)fidelity evolves to select males that will father sons that will be more successful, as occurs in classical sexual selection. Whether within-pair or extra-pair mates father more successful sons depends on the position within the cycle. The cycles occur when bourgeois males gain diminishing returns from EPC (L333). This means that bourgeois males are more effective when male help is low, which I think might cause cycling even without the inclusion of sneakers. I suggest the authors check whether cycles occur without sneakers.

In the population genetics model, I was expecting to see that the proportions of different genotypes were followed, rather than the absolute number of individuals. That is, I would remove R and Eq 5 and 6 and normalize M_i and F_j by the $\text{Sum}_i (M_i)$ and $\text{Sum}_j (F_j)$ so that they represent genotype frequencies (summing to 1). The aim of the model is to find the equilibrium frequency of sneaker males (L165). I expect that this adjustment might help to find this equilibrium frequency analytically.

The results where male help evolves to zero are superficially strange (top left of Figure 3). When $h=0$, all offspring die (L125) and both bourgeois and sneaker males spend all their time looking for mates. I gather that the population is only viable here because mutation-selection balance means $h>0$. Given the new focus on population mean fitness (Figure 1) and the 'tragedy of the commons' (L100, L230, L291), perhaps you could point out that populations may evolve to extinction when this happens. E.g., by running some simulations with lower R. In general, I also think it would be useful to point out that sneakers are effectively the same as bourgeois males but with an allele that means they always have $h=0$ (until later in the paper where sneakers have specific adaptations that make them especially effective in EPC).

Referee: 3

Comments to the Author(s).

The manuscript presents a simplified version of the previously presented model, with the removal of the sexually antagonistic component. This makes the dynamics of the model, and the manuscript as a whole, much easier to follow. I have a few additional comments on the manuscript, stated below:

Equation 1 - There is a space missing after the comma

Line 148-149 - It would be nice if δ_2 could be defined in the same way as δ_1 , so it is clear that these are comparable numbers or not. I am not sure about the meaning of "increment" in this context, specifically. It might also be worthwhile to change the numbers 1 and 2 to something more informative, e.g. "wp" for within-pair and "ep" for between pair.

Lines 123 - 124 - Are there examples of either of these cases that could be cited here? The potential mechanisms underlying these effects is not obvious.

Line 186 - "without" should be "unless"

Lines 243-244 - It would be helpful to note explicitly where the results transition from the results of the analytical model of fitness landscapes to the individual simulation results. It is clear from the figures, but would be useful to have in the text as well.

Line 272 - "is" should be "are"

Line 278 - I'm not sure what "from the same realization of simulation" means here. The same parameter combinations?

Line 302 - "can sustain" could be "can be sustained" or "can sustain itself"

Line 304 - I would suggest that the authors are explicit about this being the peak of the population fitness landscape, both here and elsewhere in the manuscript where the fitness landscape is mentioned, to avoid confusion with a sex-specific fitness landscape.

Lines 337-351 - This pattern bears a superficial resemblance to a Fisherian "sexy-sons" mechanism - namely, the increase in fitness of a mother is through the increased fitness of her sons (here, through their strategy) and their daughters tend to mate with individual who share the sons phenotype. In a Fisherian model, the advantage to sons would be solely through their mating success due to some ornament, rather than the strategy fitness benefit here. It might be worthwhile to look into the parallels between the two models.

Line 358 - Variations should be variation.

Lines 378 - Female should be females

Lines 378 - 380 - This seems like a slightly different point, as sexually antagonistic alleles are expressed, by definition, in both sexes while here you are considering the evolution of a male-specific behaviour that reduces female fitness due to lack of parental care.

Line 416 - Sperms should be sperm

Line 438 - It is not clear why the balance of the trade-off should depend specifically on the adult sex ratio. Could this be elaborated upon slightly?

Author's Response to Decision Letter for (RSPB-2020-2371.R0)

See Appendix B.

RSPB-2020-2371.R1 (Revision)

Review form: Reviewer 1

Recommendation

Accept as is

Scientific importance: Is the manuscript an original and important contribution to its field?

Good

General interest: Is the paper of sufficient general interest?

Good

Quality of the paper: Is the overall quality of the paper suitable?

Good

Is the length of the paper justified?

Yes

Should the paper be seen by a specialist statistical reviewer?

No

Do you have any concerns about statistical analyses in this paper? If so, please specify them explicitly in your report.

No

It is a condition of publication that authors make their supporting data, code and materials available - either as supplementary material or hosted in an external repository. Please rate, if applicable, the supporting data on the following criteria.

Is it accessible?

Yes

Is it clear?

Yes

Is it adequate?

Yes

Do you have any ethical concerns with this paper?

No

Comments to the Author

I think the authors have done a good job of addressing the comments raised in the previous round of review and have nothing further to add.

Decision letter (RSPB-2020-2371.R1)

04-Jan-2021

Dear Dr Li

I am pleased to inform you that your manuscript entitled "Coevolution of female fidelity and male help in populations with alternative reproductive tactics" has been accepted for publication in Proceedings B.

Open Access

Paper charges

Thank you for your excellent contribution. On behalf of the Editors of the Proceedings B, we look forward to your continued contributions to the Journal. And, finally, all the best for 2021.

Yours sincerely,
Professor Loeske Kruuk
Editor, Proceedings B
<mailto:proceedingsb@royalsociety.org>

Associate Editor:

Board Member: 1

Comments to Author:

I thank the authors for their superb job in revising this manuscript according to the comments of the referees. I believe the current version is ready for publication.

Board Member: 2
Comments to Author:
(There are no comments.)

Appendix A

Reviewer(s)' Comments to Author:

Referee: 1

Comments to the Author(s)

This manuscript presents results of an individual-based simulation model that attempts to determine how intralocus sexual conflict affects the evolution of extra-pair copulation, mediated by female fidelity and male mate guarding. The main novel result is that intralocus sexual conflict can serve to promote female fidelity and male helping behaviour since extra-pair copulation with a high-quality male results in a cost in the quality of daughters.

This is a highly interesting result, but I'm a bit concerned that the model is overly complicated. I'm not an expert on EPCs, but have no other models incorporated costs to the female? The introduction gives the impression that this is the case. Since intralocus sexual conflict is essentially a cost to the female that increases with increasing quality of the extra-pair mate, couldn't this be incorporated into an analytical framework in a relatively simple way? It's not clear to me why such a complicated model is really necessary, and it makes it difficult to compare with previous work since it includes so many different parameters, all of which could partially contribute to the observed effect. I really encourage the authors to consider building a simple analytical model that captures the main features of the simulations, to form a conceptual bridge between the simulation model and previous analytical models.

Thank you for your comments. We performed a careful literature search once more and did not find any other model that incorporated costs to the females. In the hindsight, it is probably due to two reasons. First, intuition readily suggests that if EPC brings benefits to females, they should evolve to decrease fidelity, while if EPC incurs costs for females, they should evolve to increase fidelity. A result like this would be too trivial for a model to be publishable. Second, previous work often considers high fidelity of females as the default expectation under bi-parental care, and asks what causes infidelity and EPC. This angle naturally led to a focus on the benefit, rather than the cost, of EPC. Our results also show that the population dynamics are more interesting when EPC brings benefits to the females.

As you correctly pointed out, intralocus sexual conflict indeed plays a role of introducing costs to females, but the relative cost and benefit of performing EPC for females are highly context-dependent, and this has produced the interesting results of females evolve to be highly loyal even without mate guarding under male-biased sex ratios. This result cannot be produced by simply introducing a fixed cost of EPC to females.

However, we agree with you that it would be helpful to start from a simpler case without introducing intralocus sexual conflict, male alternative reproductive strategies, and varying sex ratios at the same time. Therefore, we adopted your suggestion by replacing intralocus sexual conflict with a simpler way of introducing cost or benefit to females. We did so by assigning a survival rate of extra-pair offspring relative to within-pair offspring ($r > 0$). When $r = 1$, the extra-pair offspring has the same survival rate as the within-pair offspring; when $r > 1$, the extra-pair offspring have a higher survival rate than the within-pair offspring; when $r < 1$, the extra-pair offspring have a lower survival rate than the within-pair offspring. We further simplified the model following the suggestion of reviewer 3 by focusing on the case where the sex ratio is balanced (more relevant explanation see our reply to reviewer 3).

In the revised manuscript, we present an analytical population genetics model based on difference equations that captures the Mendelian inheritance of different male alternative reproductive strategies (sneaker and bourgeois males), but does not allow the degrees of female fidelity (u) or male help (h) to evolve. We use this model to map the fitness landscape of the population (represented by the population growth rate) under different combinations of u and h . We show that this apparently simple model can produce interesting and useful results (e.g., the fitness landscape of the population still peaks at $u = 1$ and $h = 1$ even when the extra-pair offspring have slightly higher survival rates than the within-pair offspring, see Figure 1). We then

proceeded to study the population dynamics when either h or u can evolve while the other is fixed (Figure 2), and the coevolution case where h and u evolve simultaneously (Figure 3), using individual-based simulations. We used individual-based simulations instead of the population genetics model because it captures the population dynamics more realistically (e.g. without assuming infinite population size) and can be extended easily to include other biological factors and processes (e.g. individual variations of quality, sexual selection) and different genetic architectures. Because we plan to work on extensions of the current model in a follow-up project, results of the individual based model can serve as a basis for future comparisons.

Related to the point made above, must the cost come from intralocus sexual conflict? It seems to me that other scenarios are possible which make extra-pair matings with a highly attractive male more costly. For example if he's so much in demand that the female is forced to take any opportunity to mate with him, resulting in increased predation risk. I'm sure other plausible mechanisms are possible. I think it would be interesting to investigate the effect of a general such cost not related to sexual conflict, since this would increase the generality of the result.

As explained above, we modelled the cost of female EPC differently now, via the relative survival rate of extra-pair offspring to within-pair offspring. This way of modelling the cost of benefit of EPC should increase the generality of our result. The differences in survival rates between extra-pair and within-pair offspring have been well-recognized in empirical studies, for example, in birds, this can be caused by the order of egg-laying and hatching.

Finally, are species that engage in duets and pas-de-deux usually highly sexually dimorphic? If not, then it's questionable how likely it is that intralocus sexual conflict constitutes a major cost. I think this needs to be mentioned in the discussion. All the more reason to consider a more general effect of the cost of EPCs scaling with male attractiveness, as well.

Since we adopted your suggestion of replacing intralocus sexual conflict with a more general effect of the cost of EPC (via the relative survival rate of the extra-pair offspring, r), we only briefly discuss the potential effect of intralocus sexual conflict and its implications in the Discussion.

Minor comments:

Lines 70-75: You should cite Pischedda & Chippindale 2006 here, since they directly tested this hypothesis. <https://journals.plos.org/plosbiology/article?id=10.1371/journal.pbio.0040356>

Thank you for your suggestion. Because we excluded intralocus sexual conflict from the model, the reference is no more relevant.

The figure legends to Figures 3 and 4 are pretty brief. Especially Figure 4 needs more explanation in the figure text so that the reader can navigate it without having to read the main text.

We replaced the figures 3 and 4 in the revised manuscript. And we have taken care to write more detailed explanations in the figure captions in general.

I definitely don't think that the model needs to be made more complicated, but in nature female fidelity is probably quite context-dependent. It would be nice to include a short discussion of how this might affect the outcome.

We took your suggestions of simplifying the model and discussed the context-dependency of female fidelity in the Discussion (e.g., when mate-guarding is perfectly efficient, the "intrinsic" fidelity of females may not matter anymore). Thanks again for all your insightful suggestions and comments above. We hope you will find the revised manuscript much improved after implementing the suggestions from you and the other two reviewers.

Referee: 2

Comments to the Author(s)

This study uses a mathematical model to examine the evolution of female fidelity and male parental help when there is a sexually-antagonistic trait. The main conclusion drawn from the model is that, when the sex ratio is male biased, high male helping and high female fidelity can evolve without efficient male mate guarding (L9-11, L244-248, L370-374). This cooperative outcome is thought to result from the inclusion of sexual-antagonism in the model (L214-216, L251-252, L314-319) and the authors encourage empirical consideration of intralocus sexual conflict in systems with high female fidelity (L274-276, L320-321, L374-377).

Given that intralocus sexual conflict may result from sexual selection on one sex and opposing natural selection in both sexes (L67-75), the interaction between sexual conflict and sexual selection (in this case, mate choice through EPC) is a sensible research question. A further strength of the manuscript is the incorporation of relevant biological examples (e.g., L21-34, L72-75, L132, L254-260, L283-294, L301). However, I think that the model could be analysed more comprehensively, and I give some suggestions below.

Thank you for your thoughtful and detailed comments, and for recognizing the strength and potential of our work! We thoroughly revised the manuscript taking into account the comments from you and the other two reviewers. Because we made simplifications of the model by removing intralocus sexual conflict (we replaced it with a simpler form of cost to females, namely, the reduced survival rate of extra-pair offspring) according to the suggestions of reviewer 1 and focused on the cases under balanced sex ratio as suggested by reviewer 3, some suggestions you made below could not be adopted directly any more. Instead, we paid attention to the essence of each of your comments and adjusted them to apply to our revised model. We will explain case by case in the following.

Most importantly, I recommend that the main conclusions about the importance of sexual antagonism are interrogated explicitly. I recommend re-running the model without the I_s locus to directly compare the inclusion/exclusion of intralocus sexual conflict. This is necessary to justify the conclusion that intralocus sexual conflict explains the evolution of female fidelity under low mate guarding efficiency (e.g., L209). The authors state that females are selected to produce high-fecundity daughters, which means they favour mating with males of low quality and thus choose to mate with their partner rather than engaging in EPC. However, I believe that, for these parameters (male biased sex ratios) most EPC will come from unpaired males of low quality because helping (h) evolves to be high (paired males are therefore not available for EPC, eq on L118) and pairing occurs according to quality such that unpaired males are likely to be low quality (eq on L105). Thus, it seems reasonable to hypothesise that females evolve high fidelity to avoid EPC with unpaired males of low quality, running counter the explanation in the text. In general, I would also recommend unpacking exactly how intralocus sexual conflict is hypothesised to promote female fecundity because I found it a little unclear. In particular, I wasn't sure when/why females should be selected to produce high-quality daughters (and thus prefer a low-quality male) and not high-quality sons (e.g., L75-80, L209-210, L251-252, L316-317).

The essence of this suggestion of you, we believe, is the necessity of varying only one variable at a time and use a "control treatment" to identify the effect of that variable. Since we removed the effect of intralocus sexual conflict from the current model (we do plan to study its effect further in a follow-up project), this piece of your comment cannot be applied directly. However, we took care to explore the behavior of our new model more thoroughly, for example, to always study the difference between the presence and absence of sneakers.

Male-biased sex ratios are key to the results, but the causes and likelihood of sex ratio bias are not discussed. This is crucial in establishing the relevance of the high female fidelity scenario that is outlined. In addition to a discussion, the authors could consider explicitly modelling how male-biased adult sex ratios could arise. For example, the probability of survival to reproductive age could depend on individual quality (not just parental quality and help).

Thank you for the insightful comments. In our previous model, the biased adult sex ratio was implicitly assumed to arise from condition-independent but sex-specific survival rate of juveniles after parental care has ceased. However, we agree (and also as strongly suggested by reviewer 3) that the mechanisms leading to biased adult sex ratio deserves to be studied explicitly and in more detail. Since an analysis of our basic model without biased sex ratio already revealed several interesting results (e.g., sexual conflicts leading to “tragedy of the commons”; bourgeois males “specialize” in paternal care despite high levels of cuckoldry when mate guarding is not perfectly efficient; evolutionary cycles of the frequency of sneakers, etc.) we chose to focus on balanced sex ratio in the current model, and leave the effect of biased adult sex ratio and the specific underlying mechanisms for future investigations.

I found the truncation of the Figure axes to be unsatisfying, particularly because the SI does not present the full results for regions where $\delta < 0.2$. I think that only one example is shown (Figure S3) so that it's difficult to get a sense of the overall dynamics when mate guarding is inefficient. I would imagine that $\delta = 0$ (males have no control over their partner's EPC) is a particularly important case and it is discussed on L246 and L372. To overcome the specific issue of population extinction (L180-184), perhaps the survival equation on L127 could be adjusted or, at least, justified? For example, offspring survival could be an additive function of parental quality, $s = (\alpha_f + h \cdot \alpha_m) / 2$. Changing this equation would also allow unpaired females to produce viable offspring, which might be interesting to investigate (I think this suggestion is slightly different from the issues discussed on L332-342).

Thank you for these two comments concerning the parameter range of δ and the way of modeling offspring survival. We agree that the case where mate guarding is totally inefficient ($\delta = 0$) is interesting and deserves to be included in the results. Therefore, we now included the whole parameter range of δ between 0 and 1 in the heatmaps of the revised model.

We also carefully considered your suggestion about solving the population extinction problem by allowing unpaired females to reproduce. However, we run into a different dilemma of defining “extra-pair offspring” in this case. Should we consider the offspring produced by unpaired females as extra-pair offspring? If the father of the offspring was a paired social male, then the offspring should be considered as extra-pair offspring from the father's perspective, but since the mother was never paired, it is meaningless to talk about the distinction between extra-pair and within-pair offspring for her. And since we are interested in the proportion of extra-pair offspring produced due to “female infidelity”, simply dumping the extra-pair offspring produced by paired females and all offspring from unpaired females (no matter if the unpaired females are of high or low intrinsic fidelity if they would have been paired) creates more problems than it helps to solve. Therefore, after weighing the pros and cons, we chose to solve the issue of population extinction under low male help simply by assigning a very high baseline fecundity to females so that enough offspring will survive to sustain the population even at low paternal care. Note that because of mutation-selection balance, male help would not decrease to 0 even under the “worst case scenario” in our individual based simulations.

Minor Comments:

L217-242, Figure 4, L271-272: I would suggest reducing the space dedicated to describing the case where female fidelity is subject to drift. Three paragraphs and one figure seems excessive to describe highly efficient mate guarding that prevents the female fidelity trait from being expressed. I would argue that this isn't as biologically interesting as cases where mate guarding is highly inefficient ($\delta = 0$).

Thank you for your suggestion. As you suggested, we reduced the space dedicated to explaining the population dynamics when mate guarding is fully efficient and female fidelity is subject to drift. We also put more emphasis on the cases where the efficiency of mate guarding is intermediate (when bourgeois specialize in paternal care despite of cuckoldry) or poor (the dynamics of coevolutionary cycles).

L91: I think it'd be worth considering a model where A is fixed to consider species where the frequency of sneaker males can't readily evolve. In particular, I'd be interested to know whether sneakers are necessary for the cyclical dynamics under low mate guarding efficiency that are presented in the supplementary information.

Thank you very much for the very helpful suggestion. We now included the cases where sneakers are prevented from evolving (the corresponding figures when either female fidelity of male help is fixed are provided in the Supplementary Materials Figure S12; the corresponding case under coevolution is now in the left column of Figure 3 in the main text). We will show that sneakers are indeed necessary for the evolutionary cycles and explain why it is the case in the last subsection and Figure 4 of the *Results* section.

L357-358: Although it'd be difficult to examine many different trade-offs between male help and opportunities for EPC, maybe you could examine the case where there is no trade-off? That is, where male helping does not diminish their ability to mate with another female (remove h_i from ϕ_i on L118). This would allow you to examine the evolution of female fidelity when females can always choose from all males through EPC. This may be a good test for the hypothesis that female fidelity evolves because it's beneficial to remain mated with a low-quality male partner rather than seeking high-quality males through EPC.

Thank you for your suggestion on examining the effect of individual quality variation of males on the evolution of female fidelity. In the current simplified models, we assumed the **absence** of individual variation in quality and **random** pair formation throughout. Therefore, this suggestion becomes not directly relevant for our current model. However, we will take it into account in follow-up studies, where individual variation of quality does matter.

L75-80: Given the previous result that females may prefer low quality males when selection is stronger for males than females, it seems strange that it is then assumed throughout that male competition for mating partners is stronger than female competition ($\text{Beta}=2$, L107). Perhaps the relative strength of selection on males/females is a key factor that determines whether females pursue EPC, which could be investigated by varying Beta?

Similar to your previous suggestion, since we simplified the current model by only considering random pair formation, selection on females becomes irrelevant. But thanks for the suggestion and we will pay attention to the effect of sex-specific intensity of selection in the follow-up studies where we do consider assortative pair formation.

Figs. 2 and 3: It would be easier to interpret if 'efficiency of mate guarding' was used for the x-axis on both figures rather than switching axes. I would also recommend organising the panels in the same order (rows with 'female fidelity' and 'sneakers' are switched between the two figures).

Thanks! We adopted your suggestion and now the 'efficiency of mate guarding' is always on the x-axis in all heat maps where it is relevant. We also took care to organize the panels in the same order when applicable.

L133-135, Fig. 1: why are the I_m and I_f loci included? I expect them to simply fix for the beneficial allele and then be maintained at mutation-selection balance.

In the previous model, we included I_m and I_f , the two sex-specific trait locus, to give individuals some "baseline condition" under intralocus sexual conflict. Since we include mutations, the trait values cannot literally "fix" at the sex-specific optimal, but the distribution is very peaked around the optimal values, exactly as you expected. Since we excluded intralocus sexual conflict from the current model, we also excluded these two loci.

Fig. 3: Could you clarify that 'sneakers' are counted as 'unpaired males', if correct? It might also be worth clarifying that 'male help' is 'male help and mate guarding'. The 'median female quality'

panel suggests that, in the region where there are sneaker males, the I_s locus reaches intermediate frequency. This might be an interesting case to describe in more detail.

Yes, we considered 'sneakers' as 'unpaired males' in the previous model. But also mentioned above, we now focus on balanced sex ratio at birth and as adults, so all males except the sneakers will be paired, and thus only sneakers are unpaired males in our current model.

We also made clarification of "mate help" as "male help and its by-product of mate guarding" in the revised manuscript as you suggested, in lines 125-127: "Because males that provide care almost always guard their mates to some extent, we model mate-guarding as a by-product of brood care with an efficiency $\delta \in [0,1]$."

Since individual variation in quality is now **absent** in our simplified models, we excluded the panel of 'median female quality'. But we will incorporate your suggestion in the follow-up studies where intralocus sexual conflict is present.

Figs. S1 and S2: replace 'playboys' with 'sneakers'.

Thanks for pointing out this omission of us. We took care to keep the nomenclature consistent in the revised manuscript.

The deposition of all scripts should also be commended.

Thank you very much for your compliment! We are convinced of the importance of depositing all scripts to facilitate reviewing and ensure the repeatability of our work. We continue to include all scripts of the revised manuscript in the resubmission.

We thank you again for your constructive critics. We believe that they have led to substantial improvement of the manuscript, and we hope you find our response satisfactory.

Referee: 3

Comments to the Author(s)

This is an interesting analysis on the evolution of female fidelity and male help. The authors use individual-based simulations to examine the impact of several parameters, including sex ratio biases and intra-locus sexual conflict, on the evolution of these two factors. This will be a useful addition to the literature, although I have two main concerns with the paper.

Thank you for recognizing the potential value of our previous submission of the manuscript. We address your two major concerns (related to biased sex ratio and intralocus sexual conflict) and the minor but very useful comments and suggestion in the following.

My first main concern with the paper deals with its use of biased sex ratios. The primary conclusion of this paper is that, under scenarios with a male-biased biased sex ratio, female fidelity may evolve because females prefer to stay with their current 'low-quality' mate, who will presumably produce high quality daughters due to sexual antagonism. The authors refer in the abstract to biased adult sex ratios. However, the model seems to be implementing a biased primary sex ratio – sex ratio at conception (lines 144-145 , 209-212). This is a very different scenario.

Thanks for pointing out the important difference between sex ratio at conception and sex ratio in reproducing adults. As we also explained above to reviewer 1, we made the implicit assumption of condition-independent while sex-specific mortality rate of juveniles during the life stage between the end of parental care and reproduction. This can lead to biased adult sex ratio even when the sex ratio at conception is balanced. Because mortality was assumed to be condition-

independent, this is equivalent to producing offspring directly with biased sex ratio in the individual-based model for optimizing simulation speed.

Since the following four paragraphs of your comments are tightly connected, we reply to them together.

Fisher's well-established sex ratio principle states that sex ratios are approximately 1:1 at the end of reproductive investment due to the fact that every zygote has one mother and one father. If the birth of, for example, females is less common than males, individuals who produce more females gain a disproportionate advantage, as females have a higher reproductive value relative to males (Fisher 1930 for original argument, Hamilton 1967 "Extraordinary sex ratios" or Bull and Charnov 1988 "How fundamental are Fisherian sex ratios?" section 2.2.1 for a review of the argument).

Under the assumption of equal primary sex ratios, females equally gain equally from male and female offspring. Under a biased primary sex ratio, females are transmitting more through sons, although daughters have a higher reproductive value than sons in the next generation. While the authors suggest that the evolution of the focal behaviours is due to dynamics caused by the biased adult sex ratio, I am concerned that it is in fact due to the distortion of the primary sex ratio, and the biased transmission of female genes to her offspring.

This is critical to clarify as biases in the primary sex ratios are extremely rare in systems with genetic sex determination and, under Fisherian sex ratio selection, expected to be selected against. This holds true in birds [Liker et al. 2013 "The evolution of sex roles in birds is related to adult sex ratio"]. Highly biased adult sex ratios, however, are very common in many taxa, including birds [Pipoly et al. 2015 "The genetic sex-determination system predicts adult sex ratios in tetrapods"]. Clarifying this point has a direct impact on how broadly applicable the authors' results are. Ideally, the model should be implemented with a 50/50 primary sex ratio, and then random culling can be performed to establish the adult sex ratio, which would test whether these effects are driven by the distortion of the primary sex ratio.

Generally, throughout the manuscript, the authors should be explicit about what type of sex ratio is being referred to and clarify the logic underlying their argument (In their discussion of female sex ratio adjustment, for example, the authors are referring to primary sex ratios (lines 261-262), but were just previously referring to adult sex ratios (line 247). Lines 314-317 are also based on females gaining a disproportionate fitness advantage through daughters rather than sons, and therefore on biases in the primary sex ratio. An enhanced discussion of the theory and empirical evidence for sex-ratio biases (how common are male-biased sex ratios in birds) would also be a helpful addition.

Thank you very much for your comments. We agree with the importance of implementing balanced sex ratio at conception because of the Fisher condition, and we are convinced by you and reviewer 1 about the usefulness of explicitly studying the different underlying mechanisms of biased adult sex ratios.

A thorough exploration of our revised model using the suggestions from all three reviewers, has already revealed several interesting and novel results **even at balanced adult sex ratio**, for example, 1) sexual conflict leading to "tragedy of the commons" and low population growth rates, 2) the normal (bourgeois) males evolve to provide extremely high levels of help while spending little time in pursuing extra-pair fertilization opportunities even at high levels of cuckoldry, when the efficiency of mate-guarding is intermediate, and 3) the degrees of female fidelity and male help can coevolve in evolutionary cycles when the efficiency of mate-guarding is inefficient, even at the absence of individual variation of quality and random pair formation. Therefore, we decide to focus on the case of balanced sex ratio throughout the life history in the current model, and plan to study in more detail the effect of biased adult sex ratio caused by different underlying mechanisms in a follow up study in the future.

My second main concern with the paper is the assumption that intra-locus sexual conflict is driving the observed results. This argument needs to be laid out logically, and perhaps with better justification through simulations (if there was no intra-locus sexual conflict, i.e. $a_m = a_f = 1$, would the same results be observed?). This is connected to my concerns with the sex ratio, as if female transmission is occurring mainly through males due to the biased primary sex ratio, why should they have a “purely selfish motivation (to optimize their own fitness by producing high-quality daughters)?” The authors should establish this logic more clearly, as it is the central claim of their paper.

Thank you for your comment. By considering the comments from all three reviewer, we decide to simplify the model by replacing the complex, adult sex ratio-dependent effect of EPC to females with a simpler form of cost and benefit economics in the basic model, using the relative survival rates of extra-pair offspring to within-pair offspring. As explained in detail in our reply to reviewer 1, females now receive a fixed benefit from EPC if the extra-pair offspring have higher survival rate than the within-pair offspring, and vice versa. The simplified model allows us to explore its behavior more systematically because of the reduced dimensions of parameter space.

Minor comments:

Line 37: “meta-trait” - could this be defined here?

Thank you for pointing this out. After revising the *Introduction* section to better suit our modified models, we did not include the concept of “meta-trait” in the current manuscript.

Line 78: This should be made much more clear. Females may prefer low-quality males (who will presumably produce high-quality daughters) when selection on females is high. Could the hypothesis connecting this fact to female fidelity and/or male help be explicitly stated? It’s not clear whether, for instance, females mated with high quality males will prefer EPCs under these assumptions. The connection of male help to this model is also not clear.

Since we excluded intralocus sexual conflict from the current model, we also removed this part from the Introduction. But thanks for suggesting us to state this hypothesis more explicitly. We should do so in a follow-up study where intralocus sexual conflict is included.

Lines 126-127: I think this means that the female puts in full care into the brood regardless, and the male’s amount of care is modulated by the factor h . A sentence stating that in this line to go with the expression would be helpful.

Thank you very much for your suggestion. Since we did not consider individual quality variation in females, the survival rate of offspring only depends on the amount of paternal care. We described the parameter h now in lines 120-122: “The degree of male help, $h \in [0,1]$, represents the proportion of time spent by the social male on caring for the (within- and extra-pair) offspring in his nest.”

Lines 133-134, Figure 1: I assume these are all autosomal loci, but it would be worthwhile to state this somewhere.

Yes, we only considered autosomal loci. Thank you for pointing out this important detail. Now we wrote in lines 180 “Each individual now has three evolving diploid autosomal loci, including...” and line 186 “All loci are subject to Mendelian inheritance without linkage...”.

Figure 1c: This was very helpful.

Thank you! Since our model is very much simplified now, we expect that the reader do not need an illustration of the “genetic architecture”e of the evolving traits any more.

Lines 136-139: I wonder if the authors could justify using three loci, only one of which is expressed in both sexes. This roughly corresponds to the influence of sex-specific expression on a particular trait that is nevertheless expressed in both sexes, but it would be nice to see an intuitive explanation along with the mathematical description.

Thank you for your suggestion. In the current simplified models, we assumed the absence of individual variation in trait and quality. Therefore, we only included two evolving loci, one for female fidelity and one for male help.

Line 143: “se” should be “set”. I assume the extremely high mutation rate is simply to generate enough variation for these models, but perhaps it would be useful to mention here.

Thank you for pointing out the typo!

In addition, we reduced the mutation rate to 0.01 in most of the simulations in the revised model. Indeed, as you expected, the high mutation rate was for generating enough variations and they were not necessary in most cases. We now used a high mutation rate of 0.05 only for simulating the coevolutionary cycles (for speeding up the simulation). We described the choice of mutation rates now in Lines 187-189: “Without otherwise stated, the mutation rate at each locus is set to 0.01. For the loci with continuous allelic values, the magnitude of the mutations follows a normal distribution with zero mean and a standard deviation of 0.01.”, and also in the caption of Figure 4, “We used a relatively high mutation rate of 0.05 at each locus and a mutation size distribution of zero mean and standard variation 0.05 to introduce more variations and speed up the simulations.”

Lines 147-149: It would be nice to have a description of brood mortality in relation to male help here as well.

Thanks! We revised the Models part, and the description of brood mortality in relation to male help is also added to Lines 122-125: “The survival rate of within-pair offspring is an increasing function of male care, $S(h) = \sqrt{h}$. The extra-pair offspring may have a survival advantage or disadvantage relative to their within-pair siblings. We denote the relative survival rate of extra-pair offspring as r ($r > 0$), so that their survival rate is $S(h)r$.”

Lines 206-216 – If female transmission is mainly occurring through male offspring, why are females trying to maximize the fitness of daughters? This argument should be much more clear, although see the comments on sex ratio effects above.

As mentioned above, since we removed intralocus sexual conflict, this comment becomes not immediately applicable. But thanks!

Lines 212-214: This is a confusing based on the description of how mate pairing works in this species – it was stated earlier that high quality males are more likely to pair with high quality females. Should those females then preferentially find extra-pair matings, to avoid their high-quality social mate? Given that this is the fundamental claim of the paper, it is vital that this logic is clarified and a more detailed description provided.

Thanks for identifying the ambiguity. We revised the model and only focus on the basic case of random pair formation so that individual quality variation is no more relevant.

Lines 251-252: I would like to see more justification that it is the intra-locus sexual conflict causing this result, as in lines 206-216. A better explanation of the logic underlying this claim is needed.

Again, since we removed intralocus sexual conflict from the current model, this suggestion becomes not immediately applicable. But thank you, and we will consider it in the follow-up studies, where intralocus sexual conflict will be included.

Lines 281-282: Could the authors clarify what is meant by frequent outliers here? Frequent instances of considerable male parental care?

Thank you for pointing this out. By “frequent outliers” we meant that there is a general trade of male parental care increasing with male-biased sex ratios, but there’re frequent exceptions as shown in the cited meta-analysis of Liker et al. (2015). Since we greatly simplified the models and now present a new set results, the *Discussion* section was completely rewritten, making this detailed suggestion of you not immediately relevant.

Python code: I suggest that the authors change the term “playboy” to “sneaker” to better match the descriptions in the paper, at least in the comments on the code. However, this is a very minor point.

Thank you for pointing out the inconsistency. We took care in the revised manuscript to make the nomenclature consistent throughout. We also changed the “normal males” to “bourgeois males” to be consistent with the majority of literature on male alternative reproductive tactics.

Figure S3: This is a nice illustration of the effects of finite population dynamics, as these cycles would not occur in simulations of an infinite population. Perhaps this point could be made in the main text when drift of female fidelity is discussed.

Thank you very much for this suggestion. We now included the evolutionary cycles in the main text (Figure 4 and last subsection of the *Results*), with more explanations of the dynamics.

Finally, thanks again for all of your thoughtful comments and suggestions. We have carefully considered them together with those of the other two reviewers. We believe that the manuscript is substantially improved and hope that you find our revised work satisfactory.

Appendix B

Reviewer(s)' Comments to Author:

Referee: 1

Comments to the Author(s).

This manuscript presents a model of the evolution of female fidelity and male parental care in species with alternative reproductive tactics. They find that for fixed values of female fidelity and male help, population fitness is highest at high levels of both fidelity and help. However, when these traits are allowed to evolve, the outcome changes. Female fidelity only remains high if extra-pair offspring have reduced survival, and males only care if either extra-pair offspring have reduced survival OR the efficiency of mate guarding is high. Interestingly, results from the analytical model suggest that at intermediate rates of mate guarding efficiency, males will still care despite high rates of extra-pair offspring. Results from the simulation model suggest that evolutionary cycles in male care and female fidelity can also occur.

The model and results are somewhat complicated, but the results are timely and interesting. My main criticism is that the reader gets the impression that the model seems to be set-up particularly to explain observations in systems similar to the cichlid system mentioned in the introduction. Although other systems are mentioned briefly in the discussion on lines 400-409, I felt that the generality of the results could be made more explicit by:

1. Talking more about the implications of trade-offs between guarding and caring. If parental care mainly consists of e.g. incubating eggs while the female forages or obtaining food for offspring (as in birds), then the efficiency of mate guarding while caring will be very low. How do the model results align with empirical data from systems where efficiency of mate guarding is low?

First, thank you for acknowledging the timeliness and potential impact of our work.

We appreciate the note that by setting up the model with a single example of the cichlid fish in the introduction, we might generate the impression of modeling a rather specific case. We therefore revised the introduction and included several other examples across taxa to highlight the generality of our model and results (Lines 40--45):

“Extra-pair paternity was also found in 70% of the nests of the socially monogamous beetle *Odontotaenius disjunctus*, where 54.8% of the offspring were extra-pair [8]. Many socially monogamous birds with biparental care also have surprisingly high proportions of EPO in their broods, such as black redstarts (*Phoenicurus ochruros*, 30.2% of all broods and 28.8% offspring) [9], Magellanic penguins (*Spheniscus magellanicus*, 48% of all broods and 31% of offspring) [10], and tree swallows (*Tachycineta bicolor*, 75% of all broods and 51% of offspring) [11].”

Also, following your suggestion, we included in the discussion several empirical examples where mate guarding can be of low efficiency due to diverse causes, including a case of the black coucal, where the trade-off between guarding and caring applies (Lines 393--402):

“Empirical studies demonstrated, however, mate guarding can often be inefficient due to various reasons, including: 1) female birds and mammals can often escape male paternity guarding, such as in the bluethroats (*Luscinia svecica*) [58], the yellow-breasted chats (*Icteria virens*) [59], the superb fairy-wrens (*Malurus cyaneus*) [60], and the Sika deer (*Cervus nippon*) [61]; 2) paired males may face a tradeoff between guarding and parental care, such as in black coucals (*Centropus grillii*), where parental care is provided by the males only, and once males start to incubate a (still incomplete) brood, they cannot prevent female EPC as efficiently as before, and consequently, EPO occur more often in the later-laid eggs [62]; 3) females may use stored sperm of previous mates, which is often found in insects including burying beetles [17], golden egg bugs (*Phyllomorpha laciniata*) [63], and a bee species (*Ceratina nigrolabiata*) [64].”

2. Discussing implications with respect to sexual conflict. The model is set up such that females are either willing to mate with extra pair males or not, but in many species with alternative

reproductive tactics the "sneaker" males engage in coercive mating. Is it possible to investigate this sort of scenario in your model, or would it require additional work? I'm thinking that when $r < 1$ then this could be similar to a coercive mating scenario, where mating with unpreferred coercive males has negative effects on offspring survival. I'm not sure whether the female fidelity locus can be interpreted in a sexual conflict context or not, though - maybe as females with underlying resistance traits to sneaky matings or not? At the very least, I think it would be worth bringing up whether these results are equally applicable to a sexual conflict context or not, and if so, then how.

Thank you for your suggestion! As you helpfully pointed out, we did not discuss much about the implications of our work under the framework of sexual conflict, despite that it has been one of the keywords of the entire work. We now revised the discussion by signaling more strongly where the conflict between the females and the two types of males apply, paying attention to using the keyword, for examples:

Lines 367—368: "However, sexual conflict and intrasexual competition can drive both female fidelity and male help to lower levels, and giving sneakers the opportunity to invade."

Lines 408—409: "Finally, we found that sexual conflict and the competition between male ARTs can drive the emergence of evolutionary cycles when sneakers are more competitive at extra-pair fertilization."

Lines 422—423: "Our models showed as a proof-of-principle that the coevolution of female fidelity and male help driven by the sexual conflict and male ARTs can produce interesting evolutionary dynamics."

Regarding your question about modelling coercive mating, in the current model, the sneakers are effectively the same as bourgeois males but with the allelic values at the helping locus fixed to zero ($h = 0$), until the last section where they are assigned some additional advantage in gaining fertilization opportunities. Therefore, we did not consider different survival rates of offspring produced by different types of males. Therefore, we caution to interpret the case where $r < 1$ as a consequence of unwanted coercive mating with the sneakers, because the extra-pair offspring produced with the bourgeois males will suffer from the same survival disadvantage. But this certainly would be an interesting question for future work. We discuss this in Lines 423—426:

"A valuable future extension of our model will be to include variation in individual quality. With this, we could study condition-dependent reproductive tactics, the roles of assortative mating between fecund females and attractive males, and the effect of coercive mating by unwanted (probably low quality) males."

In addition, it is a very interesting suggestion to interpret the evolution at the female fidelity locus in a sexual conflict context. Sexual conflict is most often considered as a disagreement between females and males over mating rates because of Bateman's principle. Here, however, the disagreement is no more limited to mating rate, but also **mating with whom**. Since females have the same fecundity, their fidelity directly translates to a zero-sum game between sneakers and the bourgeois males, and the outcome can be influenced by all three parties. We added the related discussion in Lines 359—363:

"We focused on analyzing the conflicts between three parties: the females, the sneaker males, and the bourgeois males. Female fidelity translates the competition between the two male ARTs into a zero-sum game within a generation, while over generations, the level of female fidelity coevolves with the level of male help and the relative frequencies of the two male ARTs."

Another less important point is that I was missing some sort of motivation why the simulations assumed that sneakers could have an exaggerated advantage, while the analytical models assumed a linear relationship. Why not incorporate this in the analytical component as well? I would have also liked some sort of prediction as to what the expected difference would be.

Thank you for your question. There might be a misunderstanding here. In fact, we only used the analytical model to map the population fitness landscape (Results section “The fitness landscape” and Figure 1). The results in Figures 2 and 3 were generated from the simulation model, where we assumed a linear relationship between the time spent on extra-pair fertilization attempts and a male’s chance of gaining extra-pair paternity. Only in the last section of the Results, “*When sneakers have an advantage in extra-pair fertilization*”, we allowed the sneakers to have an exaggerated advantage in fertilization. Thus, the assumptions in the simulation model concerning the relative fertilization advantages of bourgeois and sneakers used in Figures 2 and 3 were identical to those in the analytical model (Figure 1), and the rest of the assumptions were identical to those used in Figure 4.

Therefore, our results in Figure 4 (sneakers have an exaggerated advantage) were not directly comparable to the results of the analytical model. A more appropriate comparison (with a single variable, namely, whether sneakers have a disproportional advantage in fertilization) is to the results in Figure 3. We hope that this helps to clarify the confusion.

Related to this, reviewer 3 also suggested us to signal more strongly the transition between the results generated from the analytical model and the simulations. We do so in the following:

Line 208: “We first present the population fitness landscape generated by the analytical model”

Line 225—226: “We will show in the following with individual-based simulations...”

Line 237: “From here onwards, we present results generated from the simulation model.”

Minor comments

Line 23: change “with genetically determined two” to “with two genetically determined”

Line 98: remove “the” from “although the cooperation”

We adopted the above two suggestions. Thank you.

Line 115-116: Have you explored at all what happens when this assumption is relaxed? The manuscript already presents a lot of results, so I don’t mean you need to do this and report it, I’m just curious.

This is a very interesting question. Thanks for raising it. We indeed thought of relaxing the assumption that “only paired females can produce offspring”. However, we run into a dilemma of defining “extra-pair offspring” in this case. Should we consider the offspring produced by **unpaired** females as extra-pair offspring? If the father of the offspring was a paired social male, then the offspring should be considered as extra-pair offspring from the father’s perspective, but since the mother was never paired, it is meaningless to talk about the distinction between extra-pair and within-pair offspring for her. And since we are interested in the proportion of extra-pair offspring produced **due to “female infidelity”**, simply grouping the extra-pair offspring produced by paired females and all offspring from unpaired females (no matter whether the unpaired females are of high or low “intrinsic fidelity” if they would have been paired) also does not make sense.

Therefore, to answer your question, we need to distinguish between extra-pair offspring from the father’s perspective and those from the mother’s perspective, and also the offspring produced between an unpaired female and a sneaker/unpaired bourgeois male. It is not at all trivial to tease apart the many different routes of offspring production under the relaxed assumption. But it is certainly something very interesting to investigate in future work.

Lines 179-183: Here I think it wouldn’t hurt to be a bit more explicit, and explain that the ART locus evolves by changing frequencies while the fidelity and help loci evolve as quantitative traits.

Thanks for encouraging us to be more explicit in describing the “genetic architecture” part of our model. We adopted your suggestion and now the revised sentences are (Lines 177—180):

“The ART locus has two discrete alleles A and a , same as in the analytical model. It evolves by changing the relative frequencies of the two alleles. The other two loci have alleles that take continuous values between 0 and 1, and evolve as quantitative traits.”

Line 186: change "Without" to "Unless"
We made this correction. Thanks!

Lines 242-283: Is there any biological justification for these scenarios, or are they just carried out to better understand the dynamics when both can vary?

Thank you for the very good question! We present the results where only the level of male help **or** the level of female fidelity can evolve, not only to help the readers better understand the dynamics when both can coevolve. These cases correspond to the biological scenarios where the standing genetic variation at one of the loci is much more abundant than at the other locus. Under these situations, the evolution of one trait can be much faster than the other and can be modelled as if the male/female trait is fixed. We now added the explanations in the manuscript (Lines 237—240):

“Before studying the complete coevolutionary dynamics, we first examine the simpler cases where either the degree of male help or female fidelity is fixed, and only the other can evolve. These correspond to the biological scenarios where the standing genetic variation at one of the loci is much more abundant than at the other locus.”

Line 271: change "evolve" to "evolves"

Line 272: change "is" to "are"

Line 302: change "can sustain" to "is sustainable"

Line 348: change "sneakers (now suffer" to "sneakers (who now suffer"

We made the above four changes as you suggested. Thank you.

Line 361: I don't know if I agree that this model is particularly simple - there are very many parameters that can be varied.

You're right, it is often difficult to objectively judge whether a model is “simple” or not. We removed the word “simple” from the sentence.

We thank you again for your helpful suggestions and insightful questions. In particular, your suggestions of generalizing our results have led to, in our opinion, substantial improvement of the manuscript. We hope that you find our response satisfactory.

Referee: 2

Comments to the Author(s).

The previous version of this manuscript focused on the interaction between sexual antagonism and mate choice. The revised version does not include sexually antagonistic selection and focusses instead on the coevolution of male help and female fidelity in the presence of alternative male reproductive tactics (sneaker and bourgeois males). The key result is that males can evolve to provide a lot of parental care despite high percentages of the brood being sired by other males (specifically, by sneaker males). The authors link this result to observations of fish that care for nests containing many offspring of other males. The model simplification has made the presentation much clearer and I have only minor suggestions for improvement.

Thank you very much for acknowledging our effort and the improvement of the manuscript. Please see below our response to each of your suggestions.

The introduction says: “Theoretical models have identified... inability of females to fully compensate the loss of male care... [could cause males to] be ‘blackmailed’ to continue providing care by their paired mates, because otherwise their own genetic offspring (the ‘hostages’) will also suffer” (L47-51). This seems to be the situation described in the model – females can’t compensate for loss of male care (only females paired with bourgeois males reproduce) and males are ‘blackmailed’ into providing help for extra-pair offspring along with their own. Given this apparent precedent, I would suggest clarifying the precise novelty of the result that males provide parental care to broods with high extra-pair paternity rates.

Great suggestion! Thanks!

To produce the “females blackmailing their mates” scenario, the previous work of Kokko (1999) we cited in the manuscript requires a combination of **several conditions to present simultaneously**, including 1) initial cuckoldry frequency is low (implying that biparental care cannot evolve from a promiscuous ancestral state), 2) intrinsic cuckoldry benefits are not high, 3) females cannot fully compensate the loss of male care, and most importantly, 4) reasonably accurate detection of cuckoldry by males and the consequent reduction of parental effort in case of suspicion. If any of these conditions is violated, the state of biparental care with some relatively low levels of extra-pair paternity is no more stable, and the system evolves to polygamy where males provide no care at all.

Our model, in contrast, does not require the conditions 1), 2) and 4) above. In particular, our model predicts that at the presence of sneakers, the bourgeois males evolve to provide extremely high levels of paternal care, **no matter whether they can detect cuckoldry or not, and even if they are aware of cuckoldry, they should not decrease the amount of care**, as long as the efficiency of mate guarding is not too low. The “unconditional male care” has been identified in various taxa, for example, Dickinson (2003) showed that the western bluebird did not reduce their paternal effort even when they observed intrusion by other males and their mate accepting extra-pair copulations. In several other bird species, such as the black redstart (Villavicencio et al. 2014), the scarlet rosefinch (Schnitzer et al. 2014), and the Azure-winged magpie (Gao et al. 2020), males also did not reduce nestling provision at the loss of paternity. In the spotless starlings, the cuckolded males, surprisingly, provisioned the nestlings more than the non-cuckolded males (García-Vigón et al. 2009). In other taxa, such “unconditional male care” has been identified as well, such as in the burying beetle (Paquet et al. 2017), the plainfin midshipman fish (Bose et al. 2020), and the cichlid fish we cited in the manuscript (Bose et al. 2018). Therefore, our model contributes to the explanation the abundant cases where males provide substantial care while (possibly) being unable to accurately detect cuckoldry and/or adjusting their parental effort accordingly. A previous model by Whittingham et al. (1992) predicted a similar “full or none” way of male parental care. The model, however, did not consider the evolution of female fidelity and relied on a crucial assumption of an S-shaped relationship between offspring survival and male care, which our model does not require.

In addition, to the best of our knowledge, no pre-existing model on the coevolution of female fidelity and male help has considered male alternative reproductive tactics or imperfect mate guarding. Considering that both scenarios are widespread in nature, yet there is still no theoretical prediction about their roles on the coevolution dynamics between female fidelity and male help, we believe that our work is a timely contribution and it contains sufficient novelty to be of interest for a broad readership.

We now adopted your suggestion and highlighted more clearly the novelty of our work (Lines 50—58):

“This model requires the simultaneous presence of several conditions, including reasonably accurate detection of cuckoldry by males and the reduction of parental effort in case of suspicion. Empirical work in birds {e.g., western bluebirds (*Sialia Mexicana*) [13], black redstarts [9], scarlet rosefinches (*Carpodacus erythrinus*) [14], and Azure-winged magpies (*Cyanopica cyanus*) [15]}, fishes {e.g., the plainfin midshipman (*Porichthys notatus*) [16], and the cichlid (*Variabilichromis moori*) [7]} and the burying beetle (*Nicrophorus vespilloides*) [17], however, found that males do not seem to detect cuckoldry or react to it [for instance, in western bluebirds [13], males did not reduce paternal effort even when they observed their mate engaging in extra-pair copulations (EPCs)]. We aim to find out how such apparently maladaptive male investment can evolve.”

The model now includes a difference in survival between offspring produced from within-pair and extra-pair matings. This is important to most of the results so I would suggest that the authors add an explicit justification. I suppose that females choose high quality males when they seek out extra-pair matings and therefore have high fitness offspring. However, male quality has been removed from the model.

Another great suggestion! Indeed, we should justify the assumption of different survival rates of within- and extra-pair offspring especially at the absence of male quality variation.

Besides gaining indirect “good genes” benefit from mating with high quality males, females can often increase the heterozygosity and resultantly the survival and/or reproductive fitness of their offspring through extra-pair copulations. Most empirical support for the “genetic compatibility” benefit of female extra-pair copulations has been found in socially monogamous birds, such as the blue tit (Kempnaers et al. 1997, Foerster et al. 2003), the dark-eyed junco (Gerlach et al. 2012), and the reed bunting (Suter et al. 2007). In the Seychelles warblers, females paired with males of low MHC-diversity increased the MHC-diversity and survival of their offspring through extra-pair fertilizations (Brouwer et al. 2010). In mammals, Leclaire et al. (2013) found that the dominant female meerkats produced a higher proportion of extra-pair offspring when paired with a related male, and the extra-pair pups are also more heterozygous than the within-pair pups, although the work did not test whether the extra-pair pups also had higher survival rates.

The extra-pair offspring may not always have a survival advantage. In the coal tits, for example, no difference was found in local recruitment rate (reflecting offspring viability) between extra- and within-pair offspring (Schmoll et al. 2003, 2009). In the Magellanic penguin, the extra-pair offspring only showed slightly faster growth than the within-pair offspring, but the difference was not statistically significant (Marasco et al. 2020).

There are also cases where the extra-pair offspring suffer from lower survival than the within-pair offspring. In song sparrow, extra-pair offspring (particularly the females) were less likely to survive and become recruited to the local population, and even the recruited ones survived for significantly fewer years afterwards (Sardell et al. 2011, 2012). Similarly, in the house sparrows, extra-pair offspring have lower probability of recruiting into the breeding population (Hsu et al. 2014). The model of Lehtonen and Kokko (2015) provided a tentative explanation for female extra-pair copulations despite reduced offspring survival. They showed that if the inbreeding depression can be compensated for by inclusive fitness benefits, it can be adaptive for females to produce some less fit offspring with closely-related extra-pair males. In the saffron finches

(Saldívar et al. 2019) and the ground tits (Wang and Lu 2011), the extra-pair offspring were indeed found to be less heterozygous than the within-pair offspring.

Besides genetic compatibility, maternal effect may also affect the relative survival rates of extra- and within-pair offspring. In several bird species, extra-pair offspring are clustered in the first-laid eggs, and thus have a size and survival advantage over their within-pair half-siblings (Cordero et al. 1999, Krist et al. 2005, Magrath et al. 2009, Ferree et al. 2010). In other species, such as the black coucal, the extra-pair offspring are more likely to be the last hatchlings of the broods and suffer from higher mortality than their within-pair half-siblings (Safari and Goymann 2018).

We now provide justifications for the different survival rates of extra- and within-pair offspring in Lines 118—123:

“In the absence of individual quality variation, the different survival rates between WPO and EPO may be caused by maternal effects [34–37] or genetic compatibility. For example, in blue tits (*Cyanistes caeruleus*) [38,39], dark-eyed juncos (*Junco hyemalis*) [40] and reed buntings (*Emberiza schoeniclus*) [41], EPO survive better due to increased heterozygosity, while in song sparrows (*Melospiza melodia*) [42,43] and house sparrows (*Passer domesticus*) [44], EPO were less likely to survive, probably due to inbreeding depression [45].”

On a related note, it is important that the cyclical dynamics (Figure 3) occur when offspring from within-pair and from extra-pair matings have the same fitness ($r=1$). This means that female fidelity is not evolving due to direct fitness benefits to offspring. Instead, female (in)fidelity evolves to select males that will father sons that will be more successful, as occurs in classical sexual selection. Whether within-pair or extra-pair mates father more successful sons depends on the position within the cycle. The cycles occur when bourgeois males gain diminishing returns from EPC (L333). This means that bourgeois males are more effective when male help is low, which I think might cause cycling even without the inclusion of sneakers. I suggest the authors check whether cycles occur without sneakers.

Thank you very much for the suggestions. We first show an example of the trajectories you suggested (Figure R1) to show that the evolutionary cycles do not appear when the sneakers are absent and then we explain why it is the case.

Figure R1. The coevolutionary trajectories of female fidelity and male help when sneakers are absent. When the extra- and within-pair offspring have the same survival rate ($r = 1$) and mate guarding is not efficient ($\delta = 0$), there is no selection on female fidelity because the within- and extra-pair offspring are equally valuable for the females. Consequently, female fidelity fluctuates around the initial value ($u = 0.5$) by random drift. And since mate guarding is not efficient, the bourgeois males cannot actively influence their share of paternity and can only passively respond to the current level of fidelity of females. Therefore, the trajectory of male help follows the same pattern as the female fidelity (the bourgeois males provide relatively more help when

female fidelity happens to be relatively high and vice versa, as they balance between gaining within- and extra-pair fertilization opportunities), always at relatively low levels.

To understand why the cycles can emerge when sneakers are present but disappear when there are only bourgeois males in the population, we draw an analogy to two published models in a different coevolutionary context, namely, the coyness game proposed by Dawkins (1989). The model considers a population where females can either be “coy” or “fast” when being courted by a male. The “fast” females accept copulation immediately while the “coy” females insist an extended period of courtship during which they inspect whether the male has the characters of being a good father to their offspring. Correspondingly, there are also two types of males. The “faithful” males are willing to invest in the energetically costly and time-consuming courtship demanded by the “coy” females, while the “philanderer” males move on to search for the “fast” females. After mating, the “faithful” males stay to help the female raise the brood while the “philanderer” males desert their broods. The model assumes that females mate only once.

The model of Schuster and Sigmund (1981) studied the game using a classic evolutionary game theory approach with fixed payoffs for the female and male player under different strategy combinations and indeed, they found the following evolutionary cycles: when the “coy” females are abundant, the “philanderer” males have little chance to mate and resultantly the “faithful” males have higher fitness and increase in frequency; when most males in the population are “faithful”, the “fast” females have an advantage because they save the cost of inspecting males during the extended courtship period and therefore can start reproducing with a (most likely) faithful mate right away, so the frequency of “fast” females increase in the population; this then gives “philanderer” males the opportunity to invade; and when the frequency of “philanderer” males are high in the population, the “coy” females have higher fitness and increase in frequency, leading the cycle back to the beginning.

Later on, McNamara et al. (2008) reexamined the coyness game and pointed out that the assumption of **fixed payoff** for each sex under different strategy combinations is not realistic, and modified the model to incorporate the frequency-dependent nature of the payoffs. What they found is that under a balanced sex ratio, there are **two alternative evolutionarily stable outcomes**, namely, either all females are fast and both types of males coexist, or all females are coy and both types of males coexist. Depending on the initial condition, the population evolve to one of the two alternative ESSs and stays there forever. Since one type of female goes extinct at the ESS, there is no possibility for the system to switch between the two evolutionarily stable states. This situation is analogous to what we showed in Figure R1, where **without alternative male strategies (the sneakers) the system is trapped to a stable ESS and cannot escape**. In contrast, when sneakers are constantly present in the population (by mutations and Mendelian inheritance), the population has the opportunity to switch between different evolutionary optima as the sex-specific fitness landscape evolves [note that, like in McNamara et al. (2008), the payoffs of different male/female strategies are also frequency-dependent in our model]. The periodic switching between two ESSs (a cooperative state where females are of high fidelity and males highly helpful, and a cooperation-breakdown state where females are disloyal and males not helpful) led to the evolutionary cycles.

Now we include some additional sample trajectories when sneakers are absent and the detailed explanation of the different evolutionary dynamics in the supplementary materials section E. We also included the following explanation in the main text (Lines 345—347):

“Note that the cycles disappear when sneakers are absent (see Supplementary Materials section E for sample trajectories and explanations).”

In the population genetics model, I was expecting to see that the proportions of different genotypes were followed, rather than the absolute number of individuals. That is, I would remove R and Eq 5 and 6 and normalize M_i and F_j by the $\sum_i (M_i)$ and $\sum_j (F_j)$ so that they represent genotype frequencies (summing to 1). The aim of the model is to find the equilibrium

frequency of sneaker males (L165). I expect that this adjustment might help to find this equilibrium frequency analytically.

Thank you for your suggestion. Our analytical model has been derived from first principles, so it initially uses the absolute numbers of individuals. Normalizing M_i and F_j by dividing them by the $\text{Sum}_i (M_i)$ and $\text{Sum}_j (F_j)$ will require rewriting the whole equations in a different form (if we want to have the separate equations for frequencies). In principle, this can be done, of course, but we believe that this would not be of really much help in finding the equilibrium numbers/frequencies analytically. Note that one can use the equilibrium equations in any form to derive the equilibrium densities/frequencies. However, we found that either directly substituting the frequencies in the equilibrium equations or finding firstly equilibrium numbers and then deriving the equilibrium frequencies (by dividing M_i and F_j by dividing them by the $\text{Sum}_i (M_i)$ and $\text{Sum}_j (F_j)$) would result in long cumbersome expressions which can be solved only numerically. In addition, related to your next suggestion, keeping R in the equations without normalization can help the readers better understand the cases later in the simulation model where the population can easily go extinct when R is relatively small because of a breakdown of cooperation between males and females, when the extra-pair offspring have a survival advantage and mate guarding is sufficiently inefficient. Because of the above two reasons, we would prefer not to change the forms of Eq. 5 and 6.

The results where male help evolves to zero are superficially strange (top left of Figure 3). When $h=0$, all offspring die (L125) and both bourgeois and sneaker males spend all their time looking for mates. I gather that the population is only viable here because mutation-selection balance means $h>0$. Given the new focus on population mean fitness (Figure 1) and the ‘tragedy of the commons’ (L100, L230, L291), perhaps you could point out that populations may evolve to extinction when this happens. E.g., by running some simulations with lower R . In general, I also think it would be useful to point out that sneakers are effectively the same as bourgeois males but with an allele that means they always have $h=0$ (until later in the paper where sneakers have specific adaptations that make them especially effective in EPC).

Thank you for your suggestion. Indeed, as you suspected, the population is only viable when male help evolves towards zero but not reaching zero because of mutation-selection balance. We modified the sentences to state this more clearly (Lines 293—296):

“In the simulations, male help did not reach zero only because of mutation-selection balance, and we had to assign females very high baseline fecundity so that the population is sustainable (see supplementary Figure S16 and S17 for sample evolutionary trajectories). Under natural conditions, such populations are likely to go extinct.”

We also adopted your suggestion of pointing out the similarity between sneakers and bourgeois males when the sneakers do not have specific adaptations to extra-pair fertilization. We added this sentence (Lines 319—322):

“So far, the advantage of sneakers relative to the bourgeois in extra-pair fertilization has been proportional to the time they spend on seeking opportunities (i.e., sneakers spend full time while the bourgeois only a proportion of $1 - h$). In this sense, the sneakers were effectively the same as bourgeois with the allelic value at the h locus fixed to zero.”

In addition, to illustrate the effect of male care on the survival rate of offspring, we now provide in supplementary Figures S16—S19 four sets of evolutionary trajectories, where we show in panel (d) of each figure how the per capita growth rate of the population (the total number of survived offspring divided by the number of adults before population size culling) changes over generations, as the frequency of sneakers, female fidelity and male help coevolve. For example, in the Figure R2 (same as supplementary Figure S16) below, the extra-pair offspring have a survival advantage ($r = 1.2$), and the baseline fecundity of females was set to 100. When the efficiency of mate guarding $\delta = 0$, male help evolves to close to zero and the per capita growth rate was only about 3 at evolutionary equilibrium. This implies that the population can easily go

extinct if the baseline fecundity was set to lower numbers (this is actually how we selected the R value for the simulations).

Figure R2. Evolutionary trajectories produced for generating the pixels corresponding to $r = 1.2, \delta = 0$ on the heatmaps of Figure 3b. All trajectories were generated from a single simulation run of 20'000 generations. They were plotted in four separate panels for the clarity of illustration. The mean of the last 2000 generations of the “frequency of sneakers” trajectory in panel (a) was used for the corresponding pixel in panel “Frequency of sneakers” in Figure 3b; The mean of the last 2000 generations of the “frequency of EPO at birth” (the light green trajectory) in panel (b) was used for the panel “Frequency of EPO” in Figure 3b; the information in panel (c) was used for the panels “Degree of male help” and “Female fidelity” in Figure 3b. The per capita growth rate in panel (d) was recorded for diagnostic purposes.

Thanks again for your insightful comments and suggestions. We believe that they have led to substantial improvements of the manuscript. We hope you find our explanations and the corresponding revision of the manuscript satisfactory.

References

- Bose, A. P. H., N. Houpt, M. Rawlins, J. S. Miller, F. Juanes, and S. Balshine. 2020. Indirect cue of paternity uncertainty does not affect nest site selection or parental care in a Pacific toadfish. *Behav. Ecol. Sociobiol.* 74:24.
- Bose, A. P. H., H. Zimmermann, J. M. Henshaw, K. Fritzsche, and K. M. Sefc. 2018. Brood-tending males in a biparental fish suffer high paternity losses but rarely cuckold. *Mol. Ecol.* 27:4309–4321.
- Brouwer, L., I. Barr, M. Van De POL, T. Burke, J. A. N. Komdeur, and D. S. Richardson. 2010. MHC-dependent survival in a wild population: evidence for hidden genetic benefits gained through extra-pair fertilizations. *Mol. Ecol.* 19:3444–3455.
- Cordero, P. J., J. H. Wetton, and D. T. Parkin. 1999. Within-clutch patterns of egg viability and paternity in the house sparrow. *J. Avian Biol.* 30:103–107.

- Dawkins, R. 1989. *The selfish gene*. 40th Anniv. Oxford University Press.
- Dickinson, J. L. 2003. Male share of provisioning is not influenced by actual or apparent loss of paternity in western bluebirds. *Behav. Ecol.* 14:360–366.
- Ferree, E. D., J. Dickinson, W. Rendell, C. Stern, and S. Porter. 2010. Hatching order explains an extrapair chick advantage in western bluebirds. *Behav. Ecol.* 21:802–807.
- Foerster, K., K. Delhey, A. Johnsen, J. T. Lifjeld, and B. Kempenaers. 2003. Females increase offspring heterozygosity and fitness through extra-pair matings. *Nature* 425:714–717.
- Gao, L.-F., H.-Y. Zhang, W. Zhang, Y.-H. Sun, M.-J. Liang, and B. Du. 2020. Effects of extra-pair paternity and maternity on the provisioning strategies of the Azure-winged Magpie *Cyanopica cyanus*. *Ibis (Lond. 1859)*. 162:627–636.
- García-Vigón, E., J. P. Veiga, and P. J. Cordero. 2009. Male feeding rate and extrapair paternity in the facultatively polygynous spotless starling. *Anim. Behav.* 78:1335–1341.
- Gerlach, N. M., J. W. McGlothlin, P. G. Parker, and E. D. Ketterson. 2012. Promiscuous mating produces offspring with higher lifetime fitness. *Proc. R. Soc. B* 279:860–866.
- Hsu, Y.-H., J. Schroeder, I. Winney, T. Burke, and S. Nakagawa. 2014. Costly infidelity: low lifetime fitness of extra-pair offspring in a passerine bird. *Evolution (N. Y.)*. 68:2873–2884.
- Kempenaers, B., G. R. Verheyen, and A. A. Dhondi. 1997. Extrapair paternity in the blue tit (*Parus caeruleus*): female choice, male characteristics, and offspring quality. *Behav. Ecol.* 8:481–492.
- Krist, M., P. Nádvorník, L. Uvířová, and S. Bureš. 2005. Paternity covaries with laying and hatching order in the collared flycatcher *Ficedula albicollis*. *Behav. Ecol. Sociobiol.* 59:6–11.
- Leclaire, S., J. F. Nielsen, S. P. Sharp, and T. H. Clutton-Brock. 2013. Mating strategies in dominant meerkats: evidence for extra-pair paternity in relation to genetic relatedness between pair mates. *J. Evol. Biol.* 26:1499–1507.
- Lehtonen, J., and H. Kokko. 2015. Why inclusive fitness can make it adaptive to produce less fit extra-pair offspring. *Proc. R. Soc. B* 282:20142716.
- Magrath, M. J. L., O. Vedder, M. der Velde, and J. Komdeur. 2009. Maternal effects contribute to the superior performance of extra-pair offspring. *Curr. Biol.* 19:792–797.
- Marasco, A. C. M., J. S. Morgante, M. Barrionuevo, E. Frere, and G. P. de Mendonça Dantas. 2020. Molecular evidence of extra-pair paternity and intraspecific brood parasitism by the Magellanic Penguin (*Spheniscus magellanicus*). *J. Ornithol.* 161:125–135.
- McNamara, J. M., L. Fromhage, Z. Barta, and A. I. Houston. 2008. The optimal coyness game. *Proc. R. Soc. B* 276:953–960.
- Paquet, M., R. Wotherspoon, and P. T. Smiseth. 2017. Caring males do not respond to cues about losses in paternity in the burying beetle *Nicrophorus vespilloides*. *Anim. Behav.* 127:213–218.
- Safari, I., and W. Goymann. 2018. Certainty of paternity in two coucal species with divergent sex roles: the devil takes the hindmost. *BMC Evol. Biol.* 18:110.
- Saldívar, M. J. B., C. I. Miño, and V. Massoni. 2019. Genetic mating system, population genetics and effective size of Saffron Finches breeding in southern South America. *Genetica* 147:315–326.
- Sardell, R. J., P. Arcese, L. F. Keller, and J. M. Reid. 2012. Are there indirect fitness benefits of female extra-pair reproduction? Lifetime reproductive success of within-pair and extra-pair offspring. *Am. Nat.* 179:779–793.
- Sardell, R. J., P. Arcese, L. F. Keller, and J. M. Reid. 2011. Sex-specific differential survival of extra-pair and within-pair offspring in song sparrows, *Melospiza melodia*. *Proc. R. Soc. B* 278:3251–3259.
- Schmoll, T., V. Dietrich, W. Winkel, J. T. Epplen, and T. Lubjuhn. 2003. Long-term fitness consequences of female extra-pair matings in a socially monogamous passerine. *Proc. R. Soc. B* 270:256–264.
- Schmoll, T., F. M. Schurr, W. Winkel, J. T. Epplen, and T. Lubjuhn. 2009. Lifespan, lifetime reproductive performance and paternity loss of within-pair and extra-pair offspring in the coal tit *Periparus ater*. *Proc. R. Soc. B* 276:337–345.
- Schnitzer, J., A. Exnerová, R. Poláková, M. Vinkler, O. Tomášek, P. Munclinger, and T. Albrecht. 2014. Male ornamentation and within-pair paternity are not associated with male provisioning rates in scarlet rosefinches *Carpodacus erythrinus*. *Acta Ethol.* 17:89–97.
- Schuster, P., and K. Sigmund. 1981. Coyness, philandering and stable strategies. *Anim. Behav.* 29:186–192.
- Suter, S. M., M. Keiser, R. Feignoux, and D. R. Meyer. 2007. Reed bunting females increase fitness through extra-pair mating with genetically dissimilar males. *Proc. R. Soc. B* 274:2865–2871.
- Villavicencio, C. P., B. Apfelbeck, and W. Goymann. 2014. Parental care, loss of paternity and circulating levels of testosterone and corticosterone in a socially monogamous song bird. *Front. Zool.* 11:11.
- Wang, C., and X. I. N. Lu. 2011. Female ground tits prefer relatives as extra-pair partners: driven by kin-

selection? *Mol. Ecol.* 20:2851–2863.

Whittingham, L. A., P. D. Taylor, and R. J. Robertson. 1992. Confidence of paternity and male parental care. *Am. Nat.* 139:1115–1125.

Referee: 3

Comments to the Author(s).

The manuscript presents a simplified version of the previously presented model, with the removal of the sexually antagonistic component. This makes the dynamics of the model, and the manuscript as a whole, much easier to follow. I have a few additional comments on the manuscript, stated below:

Equation 1 - There is a space missing after the comma

Thank you very much for spotting this typo. We corrected it.

Line 148-149 – It would be nice if δ_2 could be defined in the same way as δ_1 , so it is clear that these are comparable numbers or not. I am not sure about the meaning of “increment” in this context, specifically. It might also be worthwhile to change the numbers 1 and 2 to something more informative, e.g. “wp” for within-pair and “ep” for between pair.

Thank you for your suggestion. We agree that it would be more informative to replace Δ_1 and Δ_2 with Δ_{wp} and Δ_{ep} , respectively. We have made the changes accordingly (Eqs 2,4,5,6,7 and the corresponding ones in the supplementary materials). We also replaced the word “increment” with a more informative description in the following sentences:

Line 137: “The number of survived male WPO of each genotype produced within a generation ...”;

Line 147—148: “the number of survived of female EPO of each genotype produced over a generation ...”.

Lines 123 – 124 – Are there examples of either of these cases that could be cited here? The potential mechanisms underlying these effects is not obvious.

Thank you very much for the important question. We now added the following sentences to provide empirical evidence for justifying our assumption of different survival rates of extra- and within-pair offspring (Lines 118—123):

“In the absence of individual quality variation, the different survival rates between WPO and EPO may be caused by maternal effects [34–37] or genetic compatibility. For example, in blue tits (*Cyanistes caeruleus*) [38,39], dark-eyed juncos (*Junco hyemalis*) [40] and reed buntings (*Emberiza schoeniclus*) [41], EPO survive better due to increased heterozygosity, while in song sparrows (*Melospiza melodia*) [42,43] and house sparrows (*Passer domesticus*) [44], EPO were less likely to survive, probably due to inbreeding depression [45].”

Line 186 - “without” should be “unless”

Thank you for pointing out this error. We corrected it.

Lines 243-244 – It would be helpful to note explicitly where the results transition from the results of the analytical model of fitness landscapes to the individual simulation results. It is clear from the figures, but would be useful to have in the text as well.

Thank you for your suggestion. We added the following sentence to help signal the transition:

Line 208: “We first present the population fitness landscape generated by the analytical model”

Line 225—226: “We will show in the following with individual-based simulations...”

Line 237: “From here onwards, we present results generated from the simulation model.”

Line 272 - “is” should be “are”

We made this correction. Thanks!

Line 278 - I'm not sure what "from the same realization of simulation" means here. The same parameter combinations?

Thank you for pointing out this ambiguity. We modified the corresponding sentence in the caption of Figure 3 to (Lines 310—312):

“The data at the same pixel position across heatmaps in the same panel were generated from the same simulation run (see Supplementary Figures S16 – S19 for samples of evolutionary trajectories corresponding to 4 different pixels).”

Below we use one of the newly added supplementary figures (Figure R3, same as Figure S18 of the Supplementary Materials) to illustrate how we generated the data at the pixel position ($r = 1.2, \delta = 0.6$) in the four panels of Figure 3b.

Figure R3. Evolutionary trajectories produced for generating the pixels corresponding to $r = 1.2, \delta = 0.6$ on the heatmaps of Figure 3b. All trajectories were generated from a single simulation run of 20'000 generations. They were plotted in four separate panels for the clarity of illustration. The mean of the last 2000 generations of the "frequency of sneakers" trajectory in panel (a) was used for the corresponding pixel in panel "Frequency of sneakers" in Figure 3b; The mean of the last 2000 generations of the "frequency of EPO at birth" (the light green trajectory) in panel (b) was used for the panel "Frequency of EPO" in Figure 3b; the information in panel (c) was used for the panels "Degree of male help" and "Female fidelity" in Figure 3b. The per capita growth rate in panel (d) was recorded for diagnostic purposes.

Line 302 - "can sustain" could be "can be sustained" or "can sustain itself".

Thank you for pointing out this mistake. We changed the sentence to (Line 293--296):

“In the simulations, male help did not reach zero only because of mutation-selection balance, and we had to assign females very high baseline fecundity so that the population is sustainable (see supplementary Figure S16 and S17 for sample evolutionary trajectories).”

Line 304 - I would suggest that the authors are explicit about this being the peak of the population fitness landscape, both here and elsewhere in the manuscript where the fitness landscape is mentioned, to avoid confusion with a sex-specific fitness landscape.

Great suggestion! We revised the manuscript throughout (Lines 93, 207, 208, 214, 220,223, 229, 297) to be explicit about the distinction between the peak of the population fitness landscape and the sex-specific fitness landscape.

Lines 337-351 - This pattern bears a superficial resemblance to a Fisherian “sexy-sons” mechanism – namely, the increase in fitness of a mother is through the increased fitness of her sons (here, through their strategy) and their daughters tend to mate with individual who share the sons phenotype. In a Fisherian model, the advantage to sons would be solely through their mating success due to some ornament, rather than the strategy fitness benefit here. It might be worthwhile to look into the parallels between the two models.

Another great suggestion, thank you! We now added the following sentences to draw analogy to the Fisherian “sexy sons” mechanism:

Lines 333—336: “The positive feedback resembles the Fisherian “sexy sons” mechanism in the sense that low fidelity females mate more often with sneakers (as if it was their preference) and produce “sexy (sneaker) sons” and daughters also of low fidelity (sharing the same preference for sneakers).”

Lines 342—343: “Again, the process resembles the Fisherian “sexy sons” mechanism, with the preferred male type switched to the bourgeois.”

Line 358 - Variations should be variation.

Lines 378 – Female should be females

We corrected the two typos. Thank you!

Lines 378 – 380 - This seems like a slightly different point, as sexually antagonistic alleles are expressed, by definition, in both sexes while here you are considering the evolution of a male-specific behaviour that reduces female fitness due to lack of parental care.

Indeed, thank you for pointing it out. We now revised the discussions related to the “tragedy of the commons” to make it more concise and removed this sentence.

Line 416 - Sperms should be sperm

Thanks! We made this change.

Line 438 - It is not clear why the balance of the trade-off should depend specifically on the adult sex ratio. Could this elaborated upon slightly?

Thank you for the question. In the first submission of this manuscript, we found that under male-biased adult sex ratio, females can evolve to high levels of fidelity even when mate guarding was inefficient. Our hypothesis was that the male-biased adult sex ratio has made it more valuable for females to produce highly fecund daughters, and to do so, they should pass as few as possible of the male-beneficial-female-detrimental alleles to their offspring, and to achieve this, they should avoid mating with the top-quality sneaker males. As you and the other two reviewers pointed out, despite the result being highly interesting, our analysis of the coevolutionary dynamics influenced by intra-locus sexual conflict and biased adult sex ratio was not thorough enough. For example, to produce the aforementioned result, we had to assume that females cannot adjust the sex ratio in the extra-pair offspring (against the Trivers-Willard hypothesis), and

the biased adult ratio was caused by sex-specific mortality rate of offspring independent of individual quality. Due to these limitations, we removed the components of intralocus sexual conflict and biased sex ratios in the current manuscript. Indeed, as you pointed out, the speculation we mentioned here could be misleading, and therefore, we removed this in the revised manuscript to make space for more relevant discussions suggested above by the three reviewers.

We thank you again for your insightful questions and suggestions!